# PAC Guarantees for Reinforcement Learning: Sample Complexity, Coverage, and Structure

## Abstract

Fixed–confidence (PAC) guarantees are the right primitive when data are scarce or failures are costly. This survey organizes the 2018–2025 literature through a Coverage–Structure–Objective (CSO) template, in which sample complexity satisfies $N(\varepsilon, \delta) \approx \mathsf{Cov} \times \mathsf{Comp} \times \mathrm{poly}(H) \times \varepsilon^{-2}$. Coverage captures access assumptions (online/generative vs. offline via concentrability); Structure captures problem–dependent capacity (tabular $SA$, linear dimension $d$, effective dimension $d_{\mathrm{eff}}(\lambda)$, rank $r$, Bellman/witness/BE measures); Objective fixes the target (uniform–PAC/regret, instance–dependent identification, reward–free exploration, offline control/OPE, partial observability). We synthesize: tight tabular baselines; the uniform–PAC $\Rightarrow$ high–probability regret bridge; structured learnability under Bellman rank and Bellman–Eluder dimension; linear, kernel/NTK, and low–rank models; reward–free exploration as coverage creation; and pessimistic offline RL with explicit coverage dependence. Practical outputs include a rate "cookbook," a decision tree, and a unified roadmap of open problems (kernel/NTK uniform–PAC, agnostic low–rank, misspecified offline RL, instance–dependent FA, structure selection). We unify notation, state results with explicit dependencies, and provide a decision toolkit for practitioners.

**Keywords** PAC RL; uniform–PAC; sample complexity; function approximation; offline RL; low–rank MDPs; Bellman–Eluder dimension.

**How to read this survey.** Readers mainly interested in *what to use when* can read Table 2, the CSO template (§2), and the practical decision tree (§15); then jump to the relevant setting: tabular baselines (§4), structural complexity measures (§5), function approximation (§6), rich observations/low rank (§8), reward-free exploration (§9), offline RL (§10), partial observability (§11), or PAC–Bayes (§12). Readers seeking *technical underpinnings* should read §3 and the definition blocks at the start of each section. We keep proofs informal (dependencies and scalings) and point to the primary sources for exact constants.

Figures 1 and 2, together with Tables 2 and 5, provide a quick triage guide.

## 1 Introduction

**Reinforcement learning in brief.** In reinforcement learning (RL), an agent interacts with an environment over discrete time steps. At each step, the agent observes a *state $s$* (e.g., a robot's sensor readings), selects an *action $a$* (e.g., a motor command), receives a scalar *reward $r$* (e.g., $+1$ for reaching a goal), and transitions to a new state $s'$. The agent's objective is to learn a *policy*—a rule $\pi$ mapping states to actions—that maximizes cumulative reward over a planning horizon $H$. The central challenge is the *exploration–exploitation tradeoff*: the agent must try unfamiliar actions to discover their consequences while also exploiting actions already known to be good.

This survey focuses on *fixed-confidence* (PAC) guarantees: formal bounds ensuring that, with probability at least $1 - \delta$, the learned policy is $\varepsilon$-optimal after a specified number of interactions. Such guarantees are essential when data collection is expensive or when failures are costly—settings where average-case performance is insufficient and worst-case assurances are required.

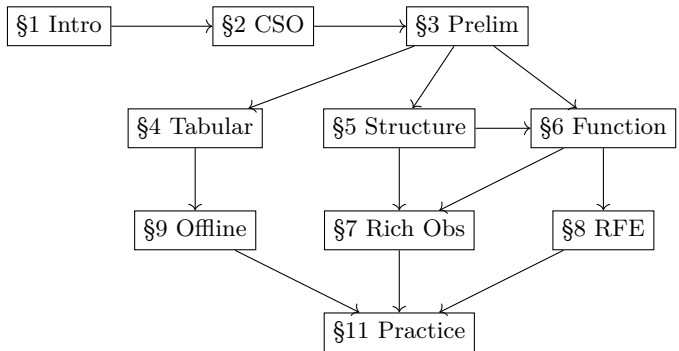

Figure 1: **Survey roadmap.** CSO framework (§2) organizes all results. Preliminaries (§3) introduce formal tools. Three pillars—tabular (§4), structural measures (§5), function approximation (§6)—feed into applications (§7–§9). Practice (§11) synthesizes. **Reading paths:** Practitioners → §2, §11, then domain sections; theorists → §3, then §4–§6.

**1. Motivation & precise scope.** This survey asks: *what fixed–confidence (PAC) guarantees are known for reinforcement learning (RL), which assumptions make them possible, and how do the guarantees scale with problem parameters?* Fixed–confidence targets are essential in safety–critical and data–scarce regimes because they bound the probability of unacceptable performance, in contrast to average regret. From 2018–2025, several strands matured: (i) *uniform–PAC* bounds that connect PAC guarantees to high–probability regret (Dann et al., 2017; (Dann et al., 2017)); (ii) *instance–dependent* identification beyond worst–case rates (Wagenmaker et al., 2022; Tirinzoni et al., 2023; (Wagenmaker et al., 2022; Tirinzoni et al., 2023)); (iii) *coverage–centric* formulations such as reward–free exploration (Jin et al., 2020; (Jin et al., 2020a)) and pessimistic offline RL (Jin–Yang–Wang, 2021; Shi et al., 2022; (Jin et al., 2021b; Shi et al., 2022b)); and (iv) *function–approximation* regimes characterized by structural measures (Jiang et al., 2017; Jin–Liu–Miryoosefi, 2021; (Jiang et al., 2017; Jin et al., 2021a)). This article synthesizes those results, emphasizing sample complexity (PAC and uniform–PAC), exploration, function approximation (linear, low–rank, kernel/over–parameterized), offline RL, partial observability, and structural risk measures. Scope is restricted to 2018–2025, with pre–2018 foundations directed to Strehl–Li–Littman (2009; (Strehl et al., 2009)), Szepesvári (2010; (Szepesvári, 2010)), Kaelbling–Littman–Moore (1996; (Kaelbling et al., 1996)), and Sutton–Barto (2018; (Sutton & Barto, 2018)).

**Scope window (2018–2025).** We focus on 2018–2025 because: (i) uniform–PAC crystallized the PAC↔regret bridge in late 2017 (Dann et al., 2017); (ii) structural complexity measures (Bellman rank, witness rank, Bellman–Eluder) matured 2017–2021 (Jiang et al., 2017; Jin et al., 2021a); and (iii) coverage–centric offline RL (pessimism, concentrability) emerged post–2020 with sharp characterizations by 2024 (Jin et al., 2021b; Shi et al., 2022b; Li et al., 2024; Zhang et al., 2024).

**2. Core concepts (informal; formalized in §3).** Let $M = (\mathcal{S}, \mathcal{A}, P, r, H, \rho)$ be a finite–horizon MDP with $|\mathcal{S}| = S$, $|\mathcal{A}| = A$, rewards in $[0, 1]$, horizon $H \in \mathbb{N}$, and start–state distribution $\rho$. For a policy $\pi$, write $V_1^\pi(\rho) = \mathbb{E}_{s \sim \rho}[V_1^\pi(s)]$.

An algorithm is $(\varepsilon, \delta)$–**PAC** if, with probability $\geq 1 - \delta$, it returns an $\varepsilon$-optimal policy $\hat{\pi}$ after $N(\varepsilon, \delta)$ episodes (Definition 1). **Uniform–PAC** bounds the number of $\varepsilon$-suboptimal episodes for *all* $\varepsilon$ simultaneously, implying high-probability regret (Definition 2, Theorem 5). Key structural parameters—Bellman rank $B$, witness rank $W$, Bellman–Eluder dimension $d_{\mathrm{BE}}$, feature dimension $d$, rank $r$—replace $(S, A)$ in sample complexity bounds (§5, §6). **Coverage** (quantified by concentrability $C_\star$) determines offline RL feasibility (Definition 6, §10).

**3. Canonical results.**

**Uniform–PAC ⇒ regret bridge (Theorem 5, informal).** If an algorithm is uniform–PAC with budget $N(\varepsilon, \delta)$, then its cumulative regret satisfies $\mathrm{Regret}(K) = \mathcal{O}\big(\int_0^H \min\{K, N(\varepsilon, \delta)\}\, d\varepsilon\big)$ with high probability.

This bridge, formalized in §3, recovers near-minimax tabular regret when $N(\varepsilon, \delta)$ has polynomial dependence in $(S, A, H, 1/\varepsilon)$ (Dann et al., 2017).

**Theorem 1** (Reward–free exploration (tabular))**.** *There exist algorithms that, without access to rewards, collect $\tilde{\mathcal{O}}\big(S^2 A H^3 \varepsilon^{-2}\big)$ episodes and subsequently, for any reward function in $[0, 1]$, output an $\varepsilon$–optimal policy with probability $1 - \delta$ (Jin et al., 2020a).*

**Theorem 2** (Linear MDPs)**.** *Under linear transition/reward realizability with feature dimension d, there exist polynomial-time algorithms with $N(\varepsilon, \delta) = \tilde{\mathcal{O}}\big(d^3 H^5 \varepsilon^{-2}\big)$ (see uniform–PAC refinements (He et al., 2021)) and $\mathrm{Regret}(K) = \tilde{\mathcal{O}}(d\sqrt{H^3 K})$. See Theorem 8 for details and constants (Jin et al., 2020b).*

**Theorem 3** (Function classes with bounded Bellman–Eluder dimension)**.** *If the Bellman–Eluder dimension $d_{\mathrm{BE}}$ of the value class is finite, then sample complexity/regret bounds scale polynomially in $(d_{\mathrm{BE}}, H)$ with logarithmic dependence on confidence (Jin–Liu–Miryoosefi, 2021; (Jin et al., 2021a)).*

**Theorem 4** (Offline RL via pessimism)**.** *In offline RL with linear MDP realizability and concentrability $C_\star$, pessimistic algorithms such as PQL attain value–estimation and control error $\varepsilon$ with data size $\tilde{\mathcal{O}}\big(\mathrm{poly}(H)\, d\, \mathrm{poly}(C_\star)\, \varepsilon^{-2}\big)$ (Shi et al., 2022; (Shi et al., 2022b)); model–based counterparts can be minimax–optimal in tabular settings (Li et al., 2024; (Li et al., 2024)).*

*Notes.* Precise constants and horizon exponents appear in the cited theorems; we state dependencies to orient the reader. Where exponents are sensitive to modeling choices, we flag refinements later.

## 4. Connections to other results.

- **Uniform–PAC as a bridge**: Uniform–PAC converts fixed–confidence guarantees into high–probability regret, aligning PAC targets with learning–curve behavior (Dann et al., 2017).

- **Coverage as a unifier**: Reward–free exploration and offline pessimism both hinge on coverage—either constructed online or inherited from a behavior policy (Jin et al., 2020a; Shi et al., 2022b).

- **Structure $\Rightarrow$ tractability**: Bellman rank and Bellman–Eluder dimension delineate when function approximation admits PAC–style learning (Jiang et al., 2017; Jin et al., 2021a).

- **Instance–dependence**: Identification can be strictly easier than worst–case learning; gap–based analyses refine $\tilde{\mathcal{O}}$ rates (Wagenmaker et al., 2022; Tirinzoni et al., 2023).

## 5. Practical implications.

- Verify realizability or adopt algorithms robust to misspecification; avoid applying linear–MDP guarantees to arbitrary deep networks without structure.

- For *multi–task* settings, consider reward–free exploration to amortize exploration across rewards (Jin et al., 2020a).

- In *offline* pipelines, quantify coverage (e.g., surrogates for $C_\star$) and prefer pessimism; avoid deployment when estimated coverage is poor (Shi et al., 2022b).

- Use policy certificates where available to obtain per–episode accountability (Dann et al., 2019; (Dann et al., 2019)).

## 6. Open problems.

i) Uniform–PAC with kernel/over–parameterized classes under minimal spectrum assumptions; target $\tilde{\mathcal{O}}(d_{\mathrm{eff}}, H, \varepsilon^{-2})$ rates with computational efficiency.

ii) Agnostic low–rank/latent–state PAC guarantees with polynomial dependence in rank/latent size and horizon.

| | Szepesvári (2010) | Strehl et al. (2009) | This survey (2018–2025) |
|---|---|---|---|
| **Venue/Type** | Monograph (Morgan & Claypool, Synthesis Lectures) | JMLR survey article | TMLR survey (2025) |
| **Scope** | Algorithms for RL; primarily tabular and basic function approximation | PAC-MDP framework; finite MDPs; sample complexity | Uniform PAC/PAC across tabular, structural measures, FA, RFE, offline, POMDP |
| **Time window** | Pre-2010 | Pre-2009 | 2018–2025 (uniform-PAC era) |
| **Primary axis** | Algorithms & theory (value iteration, TD, policy gradient) | PAC-MDP sample complexity | CSO: Coverage × Structure × Objective |
| **Guarantee types** | PAC/Regret (primarily tabular); some FA | PAC-MDP bounds for finite state-action spaces | Uniform-PAC ↔ regret; instance-dependent; offline/pessimism; PAC–Bayes |
| **Practitioner tools** | Conceptual algorithms | Theoretical foundations | Decision tree; coverage gates; certificates; rate cookbook |
| **What's new here** | N/A | N/A | Uniform-PAC synthesis; coverage-centric view; structural measures side-by-side; deployment gates |

Table 1: **Comparison with foundational surveys.** We build on Szepesvári's algorithmic treatment and Strehl et al.'s PAC-MDP framework, adding 2018–2025 advances under a unified CSO lens with deployment-oriented tools.

iii) Offline RL under model misspecification: bounds that separate estimation error, coverage, and approximation error with sharp trade–offs.

iv) Data–dependent structural selection (SRM in RL): oracle inequalities that adapt to Bellman–Eluder/Bellman rank estimated from data.

**7. Mini–bibliography (context in one–two sentences each).** Strehl–Li–Littman (2009; (Strehl et al., 2009)) surveyed PAC–MDP analyses in finite MDPs and set classical baselines for $(S, A, H, \varepsilon, \delta)$–dependent sample complexity. Szepesvári (2010; (Szepesvári, 2010)) provided a monograph–level treatment of RL algorithms and analysis. Kaelbling–Littman–Moore (1996; (Kaelbling et al., 1996)) offered an early broad survey of RL methods; Sutton–Barto (2018; (Sutton & Barto, 2018)) updated foundational algorithmics. Dann–Lattimore–Brunskill (2017; (Dann et al., 2017)) introduced uniform–PAC, linking fixed–confidence and regret in episodic RL. Jiang et al. (2017; (Jiang et al., 2017)) formalized contextual decision processes and Bellman rank; Jin–Liu–Miryoosefi (2021; (Jin et al., 2021a)) introduced Bellman–Eluder dimension. Jin et al. (2020; (Jin et al., 2020b)) established provable learning with linear function approximation; He–Zhou–Gu (2021; (He et al., 2021)) derived uniform–PAC in linear MDPs. Jin et al. (2020; (Jin et al., 2020a)) formalized reward–free exploration with near–optimal dependence on $(S, A, H, 1/\varepsilon)$. Dann et al. (2019; (Dann et al., 2019)) proposed policy certificates for accountable RL. Jin–Yang–Wang (2021; (Jin et al., 2021b)) and Shi et al. (2022; (Shi et al., 2022b)) developed pessimistic offline RL with coverage–dependent guarantees; Li et al. (2024; (Li et al., 2024)) settled model–based offline sample complexity in tabular regimes. Wagenmaker et al. (2022; (Wagenmaker et al., 2022)) and Tirinzoni et al. (2023; (Tirinzoni et al., 2023)) advanced instance–dependent PAC identification beyond worst–case rates.

**Related surveys and how we differ.** Szepesvári's monograph (Szepesvári, 2010) and Strehl–Li–Littman (Strehl et al., 2009) cover classical PAC–MDP and tabular RL. Our contribution is a *unified, book-length synthesis* across 2018–2025 that (i) connects PAC and regret via uniform–PAC across settings, (ii) places *coverage* at center stage (RFE vs. offline), (iii) systematizes structural measures (Bellman/witness/BE/bilinear) side-by-side, and (iv) provides practitioner-grade toolkits (coverage estimation, certificates) with deployment gates.

| Access & coverage | Structure (param) | Objective | Representative guarantee & refs. | Computation / oracles |
|---|---|---|---|---|
| Online | Tabular | Uniform–PAC ↔ Regret | $N = \tilde{\Theta}(SAH^3/\varepsilon^2)$; regret $\tilde{\Theta}(\sqrt{SAH^3K})$ (Domingues et al., 2021; Zhang et al., 2024; Dann et al., 2017). | Value iteration; poly-time. |
| Online | Tabular | Instance-dependent BPI | $\tilde{O}\big(\sum_{h,s} q_h(s) \sum_{a:\Delta_h(s,a)>\varepsilon} \Delta_h(s,a)^{-2}\big)$ (Wagenmaker et al., 2022; Tirinzoni et al., 2023). | UCB-style; poly-time. |
| Online (no rewards) | Tabular | Reward-free | $\tilde{O}(S^2AH^3/\varepsilon^2)$ (Jin et al., 2020a). | Model-based or CI; poly-time. |
| Online | Linear MDP ($d$) | Uniform–PAC / regret | $N = \tilde{O}(d^3H^5/\varepsilon^2)$ (to $H^4$ with refinements); regret $\tilde{O}(d\sqrt{H^3K})$ (Jin et al., 2020b; He et al., 2021). | LSVI + ridge; poly-time. |
| Online / gen. model | Kernel/NTK ($d_{\text{eff}}$) | PAC / regret | $\tilde{O}(\text{poly}(H)\,d_{\text{eff}}/\varepsilon^2)$; regret $\tilde{O}(\text{poly}(H)\sqrt{d_{\text{eff}}K})$ under completeness (Yang et al., 2020). | Kernel ridge; poly-time. |
| Online / generative | Bellman rank $B$, Witness rank $W$, BE $d_{\text{BE}}$, Bilinear $r$ | PAC / regret | $\tilde{O}(\text{poly}(H,B)\varepsilon^{-2})$, $\tilde{O}(\text{poly}(H,W)\varepsilon^{-2})$, or polynomial in $d_{\text{BE}}$, $r$ (Jiang et al., 2017; Sun et al., 2019; Jin et al., 2021a; Du et al., 2021). | Often oracle-based (CLS/CSL). |
| Online / generative | Low-rank / latent ($r$ or $m$) | PAC / exploration | $\tilde{O}(\text{poly}(H)\text{poly}(r)\varepsilon^{-2})$ (or in $m$) under identifiability/decodability (Du et al., 2019; Agarwal et al., 2020; Dann et al., 2021; Huang et al., 2023). | Rep. learning + planning. |
| Offline ($C_\star$) | Tabular / Linear | Pessimistic control; OPE | $\tilde{O}(\text{poly}(H)\,\text{poly}(\text{Comp})\,\text{poly}(C_\star)\,\varepsilon^{-2})$; tabular model-based minimax (Li et al., 2024); efficient OPE (Kallus & Uehara, 2020; Jiang & Huang, 2020). | PEVI/PQL; OPE DR/AIPW; poly-time. |
| Any | Policy class (PAC–Bayes) | Eval / selection | KL-penalized bounds for randomized policies with unbounded losses (Fard et al., 2012; Rivasplata et al., 2020; Flynn et al., 2023; Tasdighi et al., 2024). | Convex opt; poly-time. |

Table 2: Taxonomy with computational viewpoint. $\tilde{O}$ suppresses polylog factors. "CLS/CSL" = cost-sensitive classification/regression oracles.

## 2 Unified Assumption–Guarantee Framework & Taxonomy

**Axes.** We organise PAC–style results for RL along four axes and summarise each theorem as a 4–tuple (access, structure, objective, rate):

- **Access & coverage:** online interaction, generative model, or offline dataset. Coverage is quantified by coefficients (e.g., $C_\star$) that bound how well the data support the target policy.

- **Structure (complexity):** tabular; linear features ($d$); kernel/RKHS or NTK via effective dimension $d_{\text{eff}}(\lambda)$; low–rank/latent; or measures such as Bellman rank ($B$), witness rank ($W$), Bellman–Eluder dimension ($d_{\text{BE}}$), bilinear rank ($r$).

- **Objective:** uniform–PAC/regret; instance–dependent identification (BPI); reward–free exploration; offline control or OPE; partial observability; PAC–Bayes evaluation/selection.

- **Rate:** the leading sample complexity/regret dependence in $(H, \varepsilon, \delta)$ and the structural parameter(s).

**CSO template.** Most PAC results in this survey can be read through a single template:

$$N(\varepsilon, \delta) = \tilde{\mathcal{O}}\Big( \underbrace{\text{poly}(H)}_{\text{planning depth}} \cdot \underbrace{\text{Comp}(\text{structure})}_{\text{capacity}} \cdot \underbrace{\text{Cov}(\text{access/coverage})}_{\text{support}} \cdot \varepsilon^{-2} \cdot \log(1/\delta) \Big)$$

| Setting | Sample Complexity | $H$ Exponent | Tightness |
|---|---|---|---|
| Tabular (online) | $\tilde{\Theta}(SAH^3/\varepsilon^2)$ | $H^3$ | Tight (Zhang et al., 2024) |
| Linear MDP | $\tilde{O}(d^3H^5/\varepsilon^2)$ to $\tilde{O}(d^2H^4/\varepsilon^2)$ | $H^4$–$H^5$ | Upper; $H^4$ with variance reduction |
| Kernel/NTK | $\tilde{O}(d_{\text{eff}}(\lambda) \cdot H^4/\varepsilon^2)$ to $\tilde{O}(d_{\text{eff}}(\lambda) \cdot H^6/\varepsilon^2)$ | $H^4$–$H^6$ | Upper; lower bounds open |
| Low-rank $(r)$ | $\tilde{O}(\text{poly}(r) \cdot H^3/\varepsilon^2)$ to $\tilde{O}(\text{poly}(r) \cdot H^4/\varepsilon^2)$ | $H^3$–$H^4$ | Upper; depends on identifiability |
| Reward-free (tabular) | $\tilde{O}(S^2AH^3/\varepsilon^2)$ | $H^3$ | Tight (Jin et al., 2020a) |
| Reward-free (linear) | $\tilde{O}(d^2H^5/\varepsilon^2)$ | $H^5$ | Upper (Wang et al., 2020) |
| Offline (tabular) | $\tilde{\Theta}(SC_\star H^3/\varepsilon^2)$ | $H^3$ | Tight (Li et al., 2024) |
| Offline (linear) | $\tilde{O}(d \cdot C_\star \cdot H^4/\varepsilon^2)$ | $H^4$ | Upper (Shi et al., 2022b) |

Table 3: **Horizon exponents across settings.** "Tight" indicates matching upper and lower bounds (up to logs). "Upper" indicates that only upper bounds are known; optimal $H$ exponent remains open. The $H^3$ tabular exponent arises from variance accumulation over the horizon; higher exponents in function approximation settings reflect error compounding through correlated Bellman backups.

---

**Assumptions and Scope.** The guarantees in this survey hold *only when structural assumptions are satisfied*:

- **Realizability**: $Q^\star \in \mathcal{F}$ or $V^\star \in \mathcal{F}$—often violated by neural networks without careful architecture design.

- **Bellman completeness**: $\mathcal{T}_h\mathcal{F} \subseteq \mathcal{F}$—rarely verified in practice; violations cause unbounded error propagation.

- **Coverage**: Offline guarantees require bounded concentrability $C_\star < \infty$; violations cause silent failures where policies appear good on data but fail on deployment.

**Before deployment**: Use the diagnostics in §15 (Algorithms 1–2) to verify assumptions. When assumptions are uncertain, prefer pessimism and interval-valued evaluation over point policy improvement.

---

**How settings relate.** The structural settings in this survey form a hierarchy: results in more general settings recover special cases when parameters degenerate. Understanding these connections clarifies when guarantees transfer and why rates differ.

**Tabular as a special case.** Every structured setting recovers tabular MDPs under appropriate parameter choices:

- **Linear MDP** with one-hot features $(d = SA)$ recovers tabular rates $\tilde{\mathcal{O}}(SAH^3/\varepsilon^2)$.

- **Bellman rank** $B = S$ with indicator features over states.

- **Witness rank** $W = S$ when the model class contains all tabular MDPs.

- **Low-rank** with $r = S$ recovers tabular (the transition matrix has rank at most $S$).

- **Bellman–Eluder dimension** $d_{\text{BE}} = SA$ for tabular value classes.

**Inclusion hierarchy.** The settings satisfy the following (strict) inclusions, where each arrow denotes "is a special case of":

$$\text{Tabular} \subset \text{Linear MDP} \subset \text{Low-rank MDP} \subset \text{Bilinear class} \subset \text{Finite } d_{\text{BE}}$$

Each inclusion is strict: there exist problems in the larger class with infinite complexity in the smaller class. For example, continuous state spaces have $S = \infty$ but can have finite $d$ (linear) or finite $r$ (low-rank). Block MDPs with latent states $m \ll S$ are a special case of low-rank MDPs with $r = m$.

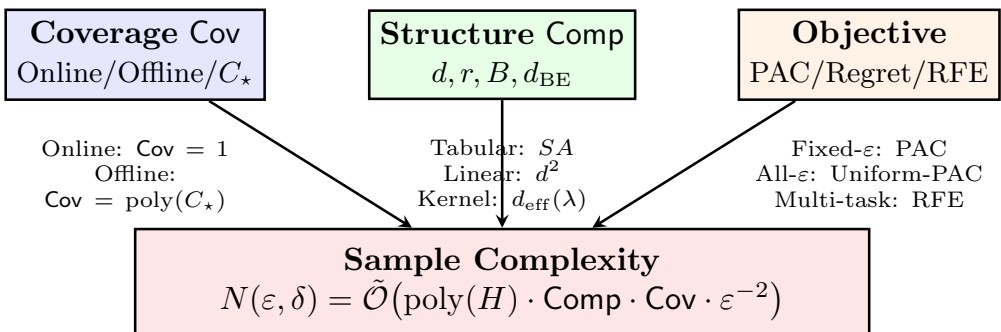

Figure 2: **CSO template: three knobs for sample complexity.** Every PAC result decomposes into Coverage × Structure × Objective. **Key insight:** the knobs are independent—improve any one to reduce $N(\varepsilon, \delta)$. **Practice:** if $N$ is large, diagnose which knob is binding: (i) coverage too weak (collect more/go online), (ii) structure too complex (better features/representation), or (iii) mismatched objective (use reward-free for multi-task). **Theory:** lower bounds hold when the other two knobs are fixed; improving one does not contradict another's lower bound.

**When do bounds match across settings?**

- Linear MDP bounds *recover* tabular bounds when $d = SA$, but with potentially worse $H$ exponents ($H^4$–$H^5$ vs. $H^3$) due to correlated estimation errors.

- Bellman rank and witness rank bounds are *incomparable*: neither subsumes the other in general, but both recover tabular with $B = W = S$.

- Bellman–Eluder dimension provides the most general guarantee but with less explicit dependence on problem parameters; it subsumes all other settings when $d_{\mathrm{BE}} < \infty$.

**Practical guidance.** When analyzing a new problem:

1. First check if the problem is tabular ($S, A$ finite and small); if so, use tabular bounds.

2. If $S$ is large but features exist, check linear realizability; use linear MDP bounds with $d \ll SA$.

3. If transitions factor through latent states, check low-rank or Block MDP structure.

4. For general function classes, compute or bound $d_{\mathrm{BE}}$ to obtain guarantees.

**Why horizon exponents differ.** In tabular settings, concentration inequalities apply independently per state-action pair, yielding $H^3$ from the product of horizon length ($H$), variance accumulation ($H$), and the union bound over stages ($H$). In linear and kernel settings, estimation errors are correlated across the feature space: a single poorly-estimated direction affects all states sharing that feature, causing errors to compound multiplicatively through Bellman backups. This yields $H^4$–$H^5$ for linear MDPs, with the gap depending on whether variance-aware estimators are used. Kernel/NTK settings inherit similar compounding, with additional dependence on spectral properties of the kernel operator. Tighter exponents require either variance-reduced estimators, instance-dependent analyses, or fundamentally different algorithmic approaches.

- **Coverage Cov (access):** $\mathsf{Cov} = 1$ online (sufficient exploration) or with a generative model; $\mathsf{Cov} = \mathrm{poly}(C_\star)$ offline; and in reward–free exploration, $\mathsf{Cov}$ is the exploration budget that *creates* reusable support.

- **Structure Comp (capacity):** tabular $\Rightarrow \Phi_{\mathrm{tab}}(S, A, H)$; linear MDPs $\Rightarrow \mathrm{poly}(d)$; kernel/NTK $\Rightarrow d_{\mathrm{eff}}(\lambda)$; Bellman/witness ranks $\Rightarrow \mathrm{poly}(B)$ or $\mathrm{poly}(W)$; BE dimension $\Rightarrow \mathrm{poly}(d_{\mathrm{BE}})$; bilinear classes $\Rightarrow \mathrm{poly}(r)$.

| Setting | Comp (structure) | Cov (coverage) | **PAC budget** $N(\varepsilon, \delta)$ **(suppressing logs)** |
|---|---|---|---|
| Tabular (online) | $\Phi_{\text{tab}}(S, A, H)$ | 1 | $\tilde{\Theta}\big(\Phi_{\text{tab}}/\varepsilon^2\big)$ |
| Linear MDP (online) | $\text{poly}(d)$ | 1 | $\tilde{\mathcal{O}}\big(\text{poly}(H)\,\text{poly}(d)\,\varepsilon^{-2}\big)$ |
| Kernel/NTK (online) | $d_{\text{eff}}(\lambda)$ | 1 | $\tilde{\mathcal{O}}\big(\text{poly}(H)\,d_{\text{eff}}(\lambda)\,\varepsilon^{-2}\big)$ |
| Bellman rank $B$ (online) | $\text{poly}(B)$ | 1 | $\tilde{\mathcal{O}}\big(\text{poly}(H, B)\,\varepsilon^{-2}\big)$ |
| Witness rank $W$ (model–based) | $\text{poly}(W)$ | 1 | $\tilde{\mathcal{O}}\big(\text{poly}(H, W)\,\varepsilon^{-2}\big)$ |
| Bilinear rank $r$ (online) | $\text{poly}(r)$ | 1 | $\tilde{\mathcal{O}}\big(\text{poly}(H, r)\,\varepsilon^{-2}\big)$ |
| Reward–free (tabular) | $\Phi_{\text{tab}}$ | exploration budget | $\tilde{\mathcal{O}}\big(S^2 A\,\text{poly}(H)\,\varepsilon^{-2}\big)$ |
| Reward–free (linear) | $\text{poly}(d)$ | exploration budget | $\tilde{\mathcal{O}}\big(\text{poly}(H)\,\text{poly}(d)\,\varepsilon^{-2}\big)$ |
| Offline (linear/tabular) | $\text{poly}(d)$ or $\Phi_{\text{tab}}$ | $\text{poly}(C_\star)$ | $\tilde{\mathcal{O}}\big(\text{poly}(H)\,\text{Comp}\,\text{poly}(C_\star)\,\varepsilon^{-2}\big)$ |

Table 4: Rate cookbook in the CSO template. Exact horizon/constant exponents appear in the cited theorems; here we surface the structural capacity and coverage factors that drive $N(\varepsilon, \delta)$.

- **Objective:** uniform–PAC/regret, instance–dependent identification, reward–free, offline control/OPE, POMDP/latent.

This *CSO* view makes explicit which knob the practitioner must turn in each setting: obtain/estimate coverage, certify structure (or use structural measures), then choose the objective and read off $N(\varepsilon, \delta)$ from the corresponding Comp and Cov.

**Decision recipe (operational).** **Step 1 (access/coverage):** Online or generative $\Rightarrow$ proceed; Offline $\Rightarrow$ estimate coverage proxies for $C_\star$ and prefer pessimism when coverage is limited. **Step 2 (structure):** Verify tabular/linear/kernel/low-rank assumptions or fall back to structural measures $(B, W, d_{\text{BE}}, r)$. **Step 3 (objective):** Choose uniform–PAC/regret, instance-dependent BPI, reward-free exploration, offline control/OPE, or partial observability. **Step 4 (verification):** Check realizability/completeness and calibrate confidence via policy certificates when available (Dann et al., 2019).

**Meta-insights from CSO factorization.**

(i) **Coverage–structure tradeoff:** Rich structure (Comp $\ll SA$) tolerates weaker coverage, but only if data span the structured subspace (e.g., rank-$r$ features must be excited).

(ii) **Objective sets $\varepsilon$-resolution:** Uniform–PAC bounds *all $\varepsilon$* simultaneously; this prevents "epsilon-cheating" and is strictly stronger than single-$\varepsilon$ PAC (Dann et al., 2017).

(iii) **Horizon compounds all factors:** Exponents vary (tabular $H^3$; linear $H^4$–$H^5$; kernel can be higher (Yang et al., 2020)). Reducing effective horizon (abstraction/discounting/hierarchy) yields superlinear savings.

## 3 Preliminaries: PAC and RL Foundations

**1. Motivation & precise scope.** This section fixes notation and formal definitions used throughout the survey and states baseline guarantees that will be referenced repeatedly. The goal is to make explicit: (i) what a Probably Approximately Correct (PAC) guarantee in RL asserts; (ii) how *uniform–PAC* strengthens PAC and connects to high–probability regret; (iii) which distributional and structural assumptions (coverage, realizability, complexity measures) underwrite finite–sample results; and (iv) how offline and reward–free settings are formalized. Clarity at this level is crucial because modern results differ primarily in their assumptions and complexity parameters rather than in proof templates. We adopt episodic finite–horizon MDPs as the default; discounted problems are mapped via an effective horizon $H_{\text{eff}} \asymp (1 - \gamma)^{-1}$ when needed. Scope covers definitions and canonical baseline theorems (informal statements with dependencies); sharp constants and refined exponents appear in later sections or in the cited papers. For historical context predating 2018 see Strehl et al. (2009); Szepesvári (2010); Kaelbling et al. (1996); Sutton & Barto (2018).

**2. Definitions.** Let $M = (\mathcal{S}, \mathcal{A}, P, r, H, \rho)$ be a finite–horizon MDP with $|\mathcal{S}| = S$, $|\mathcal{A}| = A$, horizon $H \in \mathbb{N}$, start distribution $\rho$, transitions $\{P_h\}_{h=1}^H$, and rewards $r_h \in [0, 1]$. A (possibly nonstationary) policy $\pi$ maps states to distributions over actions. Define $V_h^\pi, Q_h^\pi$ in the standard way and $V_1^\pi(\rho) = \mathbb{E}_{s \sim \rho}[V_1^\pi(s)]$.

**Definition 1** (($\varepsilon, \delta$)–PAC (fixed–confidence control)). *Alg is ($\varepsilon, \delta$)–PAC if, with probability at least $1 - \delta$, after at most $N(\varepsilon, \delta)$ interactions it outputs $\hat{\pi}$ such that $V_1^\star(\rho) - V_1^{\hat{\pi}}(\rho) \leq \varepsilon$.*

**Definition 2** (Uniform–PAC). *With probability at least $1 - \delta$, for all $\varepsilon > 0$ simultaneously the number of $\varepsilon$–suboptimal episodes is at most $N(\varepsilon, \delta)$. Uniform–PAC implies a high–probability regret bound by summing over $\varepsilon$ levels (Dann et al., 2017).*

**Definition 3** (Function class; realizability; agnostic). *Let $\mathcal{F}$ be a value or $Q$–function class. Realizability means $Q^\star \in \mathcal{F}$ (or $V^\star \in \mathcal{F}$); the agnostic setting allows $Q^\star \notin \mathcal{F}$ and measures error relative to $\inf_{f \in \mathcal{F}} \|f - Q^\star\|$.*

**Definition 4** (Covering number). *$\mathcal{N}(\epsilon, \mathcal{F}, \|\cdot\|)$ is the size of the smallest $\epsilon$–net of $\mathcal{F}$ under norm $\|\cdot\|$.*

**Definition 5** (Bellman operator and error). *$(\mathcal{T}_h f)(s, a) = r_h(s, a) + \mathbb{E}_{s' \sim P_h(\cdot|s,a)} \max_{a'} f_{h+1}(s', a')$. The stagewise Bellman error is $\|f_h - \mathcal{T}_h f\|$ in the specified norm.*

**Definition 6** (Occupancy and concentrability). *Let $d_h^\pi(s, a)$ be the occupancy of policy $\pi$ at stage $h$ and $\mu_h(s, a)$ the data (behavior) occupancy. Define $C_\star = \max_h \left\| \frac{d_h^\pi}{\mu_h} \right\|_\infty = \max_{h,(s,a)} \frac{d_h^\pi(s,a)}{\mu_h(s,a)}$.*

**Definition 7** (Access models). *Online: sequential interaction; generative model: i.i.d. sampling from $P_h(\cdot \mid s, a)$ on query $(s, a, h)$; offline: fixed dataset $\mathcal{D}$ with no further interaction.*

**Definition 8** (Observation/context space). *In settings with rich observations (§5, §8), let $\mathcal{X}$ denote the observation or context space. For standard MDPs, $\mathcal{X} = \mathcal{S} \times \mathcal{A}$; for contextual decision processes (CDPs), $\mathcal{X}$ may be high-dimensional (e.g., images) while latent states $z \in [m]$ with $m \ll |\mathcal{X}|$ govern dynamics.*

**Definition 9** (Tabular complexity shorthand). *We write $\Phi_{\mathrm{tab}}(S, A, H) \equiv S A H^3$ to denote the canonical tabular factor in episodic sample complexity/regret bounds, up to absolute constants and logarithms.*

## 3. Canonical results (informal statements with dependencies).

**Theorem 5** (Uniform–PAC $\Rightarrow$ high–probability regret; tabular). *If an algorithm is uniform–PAC with budget $N(\varepsilon, \delta)$, then with probability $1 - \delta$ its cumulative regret after $K$ episodes satisfies*

$$\mathrm{Regret}(K) = \mathcal{O}\Big( \int_0^H \min\{K, N(\varepsilon, \delta)\} \, d\varepsilon \Big),$$

*which recovers near–minimax tabular regret rates when $N(\varepsilon, \delta)$ has polynomial dependence in $(S, A, H, 1/\varepsilon)$ (Dann et al., 2017).*

**Tightness:** The conversion is tight; uniform-PAC is strictly stronger than single-$\varepsilon$ PAC, and the regret integral cannot be improved in general (Dann et al., 2017).

**Theorem 6** (Tabular minimax sample complexity is $\tilde{\Theta}(SAH^3/\varepsilon^2)$). *In finite-horizon tabular MDPs with $S$ states, $A$ actions and horizon $H$, the optimal (worst-case) online sample complexity to learn an $(\varepsilon, \delta)$-optimal policy is*

$$N(\varepsilon, \delta) = \tilde{\Theta}\Big( \frac{SAH^3}{\varepsilon^2} \Big),$$

*and the minimax regret after $K$ episodes is $\tilde{\Theta}\big( \min\{\sqrt{SAH^3 K}, HK\} \big)$, both up to polylogarithms in $(S, A, H, 1/\varepsilon, 1/\delta)$. These scalings are achieved by optimistic value-iteration variants and are minimax-sharp up to log factors; see Zhang et al. (2024) for tight upper/lower bounds and Domingues et al. (2021) for refined lower bounds.*

**Tightness:** Tight ($\Theta$); matching upper and lower bounds up to logarithms (Zhang et al., 2024; Domingues et al., 2021).

**Theorem 7** (Reward–free exploration (tabular)). *There exist algorithms that, without accessing rewards, collect*

$$N_{\mathrm{RFE}}(\varepsilon, \delta) = \tilde{\mathcal{O}}\Big( \frac{S^2 A H^3}{\varepsilon^2} \log \frac{1}{\delta} \Big)$$

*episodes and subsequently, for any reward $r \in [0,1]$, compute an $\varepsilon$–optimal policy w.p. $\geq 1 - \delta$. The $S^2$ factor (vs. $S$ in online learning) comes from covering occupancies for* all *downstream rewards (Jin et al., 2020a).*

**Tightness:** Tight; matching lower bound shows $S^2$ factor is necessary (Jin et al., 2020a).

**Theorem 8** (Linear MDPs; realizable function approximation)**.** *Under linear transition/reward realizability with feature dimension d, LSVI–UCB achieves*

$$N(\varepsilon, \delta) \;=\; \tilde{\mathcal{O}}\Big(\frac{d^3 H^5}{\varepsilon^2} \log \frac{1}{\delta}\Big) \quad and \quad \mathrm{Regret}(K) \;=\; \tilde{\mathcal{O}}\big(d\sqrt{H^3 K}\big).$$

*Refinements improve the horizon dependence to $H^4$ via sharper concentration. Uniform–PAC analogues hold under Bellman completeness (He et al., 2021).*

**Tightness:** Upper bound only. The $H^5$ exponent can be reduced to $H^4$ with variance-aware estimators; optimal $H$ exponent remains open. Dependence on $d$ is believed tight up to polynomial factors.

**Theorem 9** (Function classes with bounded Bellman–Eluder dimension)**.** *If the Bellman–Eluder dimension $d_{\mathrm{BE}}$ of the value class is finite, sample–efficient learning is possible with rates polynomial in $(H, d_{\mathrm{BE}})$ (precise exponents depend on the algorithmic template) (Jin et al., 2021a).*

**Tightness:** Upper bound only. Provides general learnability guarantee; lower bounds for specific $d_{\mathrm{BE}}$ values are known only in special cases.

**Theorem 10** (Offline RL via pessimism; linear MDPs (control))**.** *In offline RL with linear realizability and concentrability $C_\star$, pessimistic algorithms (e.g., PQL) achieve control error $\varepsilon$ with dataset size $\tilde{\mathcal{O}}\big(\mathrm{poly}(H)\,\mathrm{poly}(d)\,\mathrm{poly}(C_\star)\,\varepsilon^{-2}\big)$ (Shi et al., 2022b); in tabular model–based settings, minimax–optimal sample complexity is attained (Li et al., 2024).*

**Tightness:** Upper bound only for linear setting. Tabular model-based offline RL is minimax-tight (Li et al., 2024); linear lower bounds with matching $C_\star$ dependence remain open.

## 4. Connections to other results.

- **Uniform–PAC as a bridge:** converts fixed–confidence statements into regret bounds and supports per–episode guarantees (Dann et al., 2017).

- **Coverage as currency:** reward–free exploration *creates* coverage online, while offline pessimism *consumes* coverage quantified by $C_\star$ (Jin et al., 2020a; Shi et al., 2022b).

- **Structure $\Rightarrow$ learnability:** Bellman rank and Bellman–Eluder dimension delineate when function approximation admits PAC–style guarantees (Jiang et al., 2017; Jin et al., 2021a).

- **From worst–case to instance–dependent:** identification and gap–based analyses refine tabular rates and motivate adaptive exploration (see Section 4).

## 5. Practical implications.

- Check assumptions before invoking a bound: realizability ($Q^\star \in \mathcal{F}$), coverage ($C_\star$), and horizon scaling; do not transfer linear–MDP guarantees to arbitrary deep networks.

- For multi–reward or task families, use reward–free exploration to amortize exploration cost across rewards (Jin et al., 2020a).

- In offline pipelines, estimate coverage proxies and prefer pessimistic objectives; avoid deployment when coverage is poor (Shi et al., 2022b).

- When available, compute policy certificates to obtain per–episode accountability (Dann et al., 2019).

**6. Open problems.**

i) Uniform–PAC under kernel/over–parameterized classes with minimal spectral assumptions and near–optimal $(H, d_{\text{eff}}, \varepsilon^{-2})$ dependence.

ii) Agnostic PAC guarantees for low–rank/latent models with polynomial dependence on rank/latent size and horizon.

iii) Offline RL under misspecification: bounds separating approximation, estimation, and coverage with sharp trade–offs in $(H, d, C_\star)$.

iv) Data–driven structural selection (SRM for RL): oracle inequalities that adapt to Bellman–Eluder/Bellman rank estimated from trajectories.

**7. Related work.** **Strehl–Li–Littman (2009)** surveyed PAC–MDP analyses in finite MDPs and set classical $(S, A, H, \varepsilon, \delta)$ baselines (Strehl et al., 2009). **Szepesvári (2010)** provided a monograph–level treatment of RL algorithms and analysis (Szepesvári, 2010). **Kaelbling–Littman–Moore (1996)** offered an early broad survey; **Sutton–Barto (2018)** updated foundational algorithmics (Kaelbling et al., 1996; Sutton & Barto, 2018). **Dann–Lattimore–Brunskill (2017)** introduced uniform–PAC and showed how it yields high–probability regret bounds (Dann et al., 2017). **Domingues et al. (2021)** developed tight lower/upper bounds for episodic tabular RL (Domingues et al., 2021); **Zhang et al. (2024)** gave a sharp characterization of optimal online sample complexity (Zhang et al., 2024). **Jiang et al. (2017)** formalized Bellman rank and oracle–efficient learning in contextual decision processes (Jiang et al., 2017); **Jin–Liu–Miryoosefi (2021)** introduced Bellman–Eluder dimension for function approximation (Jin et al., 2021a). **Jin et al. (2020)** and **Wang et al. (2021)** established learning with linear features and near–optimal regret/PAC dependencies; **He–Zhou–Gu (2021)** derived uniform–PAC for linear MDPs (Jin et al., 2020b; He et al., 2021). **Jin et al. (2020)** formalized reward–free exploration with near–optimal dependence on $(S, A, H, 1/\varepsilon)$ (Jin et al., 2020a). **Shi et al. (2022)** analyzed pessimistic offline RL with concentrability dependence, and **Li et al. (2024)** settled tabular model–based offline complexity (Shi et al., 2022b; Li et al., 2024).

# 4 Tabular Baselines: Minimax and Instance-Dependent PAC

**1. Motivation & precise scope.** This section establishes the finite-sample landscape for tabular episodic RL, the setting in which the sharpest PAC and uniform-PAC guarantees are known. The central questions are: (i) what are the *minimax* sample-complexity rates to obtain an $\varepsilon$–optimal policy with confidence $1 - \delta$; and (ii) when can *instance-dependent* structure yield substantially faster identification than worst-case rates. The answers calibrate what is statistically achievable without function approximation and serve as baselines for structured models in later sections. We present lower/upper bounds with their dependencies on $(S, A, H, \varepsilon, \delta)$, summarize best-policy identification (BPI) guarantees that depend on suboptimality gaps and reachability, and explain how uniform-PAC connects these fixed-confidence results to regret. We also situate per-episode *policy certificates* that furnish auditable guarantees during learning. Throughout, constants and horizon exponents follow the cited theorems and we defer exact exponents to the cited theorems.

**2. Definitions.** We work in the finite-horizon tabular MDP $M = (\mathcal{S}, \mathcal{A}, \{P_h\}_{h=1}^H, \{r_h\}_{h=1}^H, H, \rho)$ with $|\mathcal{S}| = S$, $|\mathcal{A}| = A$, rewards $r_h \in [0, 1]$, and start distribution $\rho$. Let $V_h^\pi, Q_h^\pi$ be the value and action-value functions of a (possibly nonstationary) policy $\pi$, and let $(V^\star, Q^\star)$ denote optimal values.

**Definition 10** (($\varepsilon, \delta$)–PAC)**.** *An algorithm is $(\varepsilon, \delta)$–PAC if, with probability at least $1 - \delta$, after at most $N(\varepsilon, \delta)$ episodes it outputs $\hat\pi$ such that $\mathbb{E}_{s \sim \rho}\left[V_1^\star(s) - V_1^{\hat\pi}(s)\right] \leq \varepsilon$.*

**Definition 11** (Uniform–PAC)**.** *An algorithm is uniform–PAC if, with probability at least $1 - \delta$, for all $\varepsilon > 0$ simultaneously the number of $\varepsilon$–suboptimal episodes is at most $N(\varepsilon, \delta)$; summing the per-episode suboptimality yields a high-probability regret bound (Dann et al., 2017).*

**Definition 12** (Best-Policy Identification (BPI), gaps, reachability)**.** *For each stage $h$ and state $s$, define the* action gap $\Delta_h(s, a) := Q_h^\star(s, a_h^\star(s)) - Q_h^\star(s, a)$ *with* $a_h^\star(s) \in \arg\max_a Q_h^\star(s, a)$*. Let $d_h^\pi(s)$ be the state occupancy under $\pi$ at stage $h$ and $q_h(s) := \sup_\pi d_h^\pi(s)$ a reachability factor. BPI asks for an algorithm that*

*outputs a policy $\hat{\pi}$ with $\mathbb{E}_{s\sim\rho}[V_1^\star(s) - V_1^{\hat{\pi}}(s)] \leq \varepsilon$ with probability $1 - \delta$ and whose sample complexity scales with $\{\Delta_h(s,a), q_h(s)\}$ rather than $(S, A)$ alone.*

**Definition 13** (Policy certificate). *A policy certificate at episode $t$ is a data-dependent upper bound $U_t$ such that, with probability at least $1 - \delta$, $V_1^\star(\rho) - V_1^{\pi_t}(\rho) \leq U_t$; certificates enable per-episode accountability (Dann et al., 2019).*

## 3. Canonical results (informal statements with conditions).

**Theorem 11** (Minimax tabular sample complexity). *In finite-horizon tabular MDPs with $S$ states, $A$ actions, and horizon $H$, the worst-case sample complexity for learning an $(\varepsilon, \delta)$-optimal policy is*

$$N(\varepsilon, \delta) \;=\; \tilde{\Theta}\Big(\frac{S\,A\,H^3}{\varepsilon^2}\Big),$$

*and the minimax regret after $K$ episodes is $\tilde{\Theta}\big(\min\{\sqrt{SAH^3K}, HK\}\big)$, both up to logarithmic factors. See Domingues et al. (2021); Dann et al. (2017); Zhang et al. (2024) for sharp lower/upper bounds and exact constants.*

**Uniform–PAC $\Rightarrow$ regret.** By Theorem 5, uniform–PAC budgets directly imply high-probability regret bounds: integrating $\min\{K, N(\varepsilon, \delta)\}$ over $\varepsilon$ recovers near-minimax tabular regret $\tilde{\mathcal{O}}(\sqrt{SAH^3K})$ when $N(\varepsilon, \delta) = \tilde{\mathcal{O}}(SAH^3/\varepsilon^2)$.

**Theorem 12** (Instance-dependent identification (BPI)). *There exist optimistic identification algorithms whose sample complexity satisfies*

$$N_{\mathrm{BPI}}(\varepsilon, \delta) \;=\; \tilde{\mathcal{O}}\left(\sum_{h=1}^{H}\sum_{s\in\mathcal{S}} q_h(s) \sum_{\substack{a\in\mathcal{A}: \\ \Delta_h(s,a)>\varepsilon}} \frac{1}{\Delta_h(s,a)^2}\right),$$

*and lower bounds showing that dependence on inverse squared gaps and reachability is unavoidable up to logarithms (Wagenmaker et al., 2022; Tirinzoni et al., 2023). Precise constants and logarithmic factors appear in Wagenmaker et al. (2022); Tirinzoni et al. (2023).*

**Theorem 13** (Policy certificates). *For tabular episodic MDPs there exist algorithms that compute per-episode policy certificates $U_t$ such that, with probability at least $1 - \delta$, $V_1^\star(\rho) - V_1^{\pi_t}(\rho) \leq U_t$ and the cumulative sum of certificates tracks the algorithm's regret up to lower-order terms (Dann et al., 2019).*

## 4. Why tabular matters: CSO lens.

- **CSO instantiation:** $\mathsf{Comp} = SAH^3$; $\mathsf{Cov} = 1$ (online); $\mathsf{Obj}$ varies (uniform–PAC, BPI, RFE).

- **Calibration role:** Any structured rate (linear, low-rank) must *recover* tabular when structure degenerates (e.g., one-hot features with $d = SA$). Theorem 11 sets the ceiling.

- **Uniform–PAC bridge:** Theorem 5 shows PAC $\to$ high-probability regret by integrating counts of $\varepsilon$-bad episodes (Dann et al., 2017).

- **Instance-dependence:** BPI (Theorem 12) shrinks $\mathsf{Comp}$ from $SA$ to a gap- and reachability-weighted sum when $\Delta_h(s,a)$ are large on reachable states.

## 5. Practical implications.

- Use algorithms with explicit confidence sets (optimism or posterior bonuses) to ensure uniform–PAC behavior; avoid heuristics without quantifiable uncertainty.

- When identification is the goal, prioritize exploration toward states with high estimated reachability and ambiguous gaps; avoid wasting samples where $\Delta_h(s,a)$ is confidently large.

- Track policy certificates during training and gate deployment on $U_t \leq \varepsilon$; if certificates stall above threshold, collect more data or revise exploration.

- Treat all PAC claims as conditional on the tabular assumption; do not extrapolate rates to function approximation without verifying realizability and structure.

**5b. Mini-example: diagnosing linear realizability.** **Setting:** CartPole with state $(x, \dot{x}, \theta, \dot{\theta})$ and actions {left, right}. Candidate features $\phi(s, a) = [x, \dot{x}, \theta, \dot{\theta}, \mathbf{1}\{a = \text{right}\}]^\top$.

**Protocol:** (a) collect $n = 500$ random-policy transitions; (b) for each $h$, fit $\hat{Q}_h = \phi^\top w_h$ via ridge; (c) compute held-out Bellman residuals $\mathcal{E}_h = |\hat{Q}_h - (r_h + \max_{a'} \hat{Q}_{h+1}(s', a'))|$; (d) **pass** if $\frac{1}{n} \sum_i \mathcal{E}_h^2 \lesssim \varepsilon^2$ for target $\varepsilon$. **Failure signal:** residuals grow with $h$ or show bias near $\theta \approx \pm \pi/2$. **Remedy:** add interactions $[\phi, \phi \odot \phi]$ or switch to kernels $\kappa(s, s') = \exp(-\|s - s'\|^2/\sigma^2)$.

**5c. Failure mode: violated Bellman completeness.** **Scenario:** $10 \times 10$ GridWorld with walls; $\phi(s, a) = [x, y, a]^\top$. Wall "bounce" makes $\mathbb{E}[V_{h+1}(s')|s, a]$ nonlinear. **Symptom:** LSVI-UCB converges suboptimally; residuals plateau $\approx 0.3$ for target $\varepsilon = 0.01$; near-wall estimates highly variable. **Root cause:** linear realizability/completeness fails. **Fix:** (i) augment with wall features (adjacent, corner, distance); (ii) use model-based/witness-rank tests; or (iii) accept agnostic error $\varepsilon_{\text{approx}}$ and aim for bounds that degrade gracefully. **Consequence:** Linear guarantees (Theorem 19) do not apply; rates can revert toward tabular $\tilde{\Theta}(SAH^3/\varepsilon^2)$ absent structure.

**6. Open problems.**

i) Determine the sharp *finite* constants and exact horizon exponents for the minimax PAC sample complexity in episodic tabular MDPs; validate tightness via matching lower bounds (see Zhang et al. (2024) for exact exponents)

ii) Design identification algorithms whose instance-dependent rates match lower bounds simultaneously for all gaps $\{\Delta_h(s, a)\}$ and all reachability profiles $\{q_h(s))\}$ (up to logarithms).

iii) Develop policy-certificate schemes with near-linear-time updates per episode and provable calibration under nonstationary rewards or transitions.

iv) Establish robust uniform–PAC guarantees under small model misspecification in rewards or transitions (e.g., bounded KL or TV perturbations).

**7. Related work.** **Domingues et al. (2021)** derived refined lower and upper bounds for episodic tabular RL, tightening the worst-case sample complexity landscape (Domingues et al., 2021). **Zhang et al. (2024)** settled the optimal online sample complexity for tabular RL, clarifying horizon and $(S, A)$ dependences to which PAC upper bounds should align (Zhang et al., 2024). **Dann, Lattimore, Brunskill (2017)** introduced uniform–PAC and showed how it yields high-probability regret bounds, forging a bridge between PAC and regret analyses (Dann et al., 2017). **Wagenmaker, Simchowitz, Jamieson (2022)** established instance-dependent identification rates and corresponding lower bounds, revealing regimes where identification is strictly easier than worst-case learning (Wagenmaker et al., 2022). **Tirinzoni, Al-Marjani, Kaufmann (2023)** provided optimistic PAC algorithms with instance-dependent guarantees and clarified the role of reachability in BPI (Tirinzoni et al., 2023). **Dann et al. (2019)** introduced policy certificates enabling per-episode accountability and linked certificate sums to regret (Dann et al., 2019). **Jin et al. (2020)** formulated reward-free exploration with near-optimal tabular rates, highlighting the role of exploration-driven coverage for downstream control (Jin et al., 2020a).

## 5 Structural Complexity for RL

**1. Motivation & precise scope.** This section formalizes the structural parameters that characterize statistical learnability of reinforcement learning with rich observations and function approximation. The

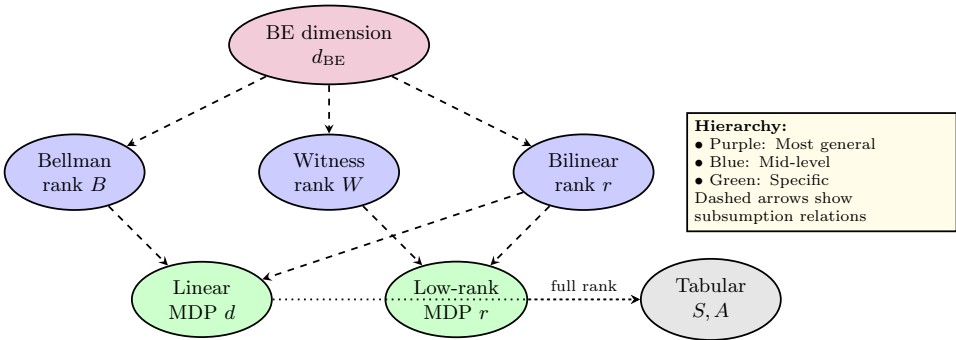

Figure 3: Hierarchy of structural complexity measures. Bellman–Eluder dimension is most general; specific settings are special cases. Tabular RL recovers via one-hot features or full-rank factorizations.

| Measure | Assumption | Algorithms | Rate prototype | Notes |
|---|---|---|---|---|
| Bellman rank $B$ | CDP with rank factorization | OLIVE / elimination | $\tilde{\mathcal{O}}(\text{poly}(H,B)\,\varepsilon^{-2})$ | Model-free; needs realizability (Jiang et al., 2017). |
| Witness rank $W$ | Model class & discriminator | Model-based optimism | $\tilde{\mathcal{O}}(\text{poly}(H,W)\,\varepsilon^{-2})$ | Witness tests rule out models (Sun et al., 2019). |
| BE dim. $d_{\text{BE}}$ | Value-class info complexity | BE-optimistic | $\tilde{\mathcal{O}}(\text{poly}(H,d_{\text{BE}})\,\varepsilon^{-2})$ | Info-theoretic; general (Jin et al., 2021a). |
| Linear MDP ($d$) | Linear realizability/completeness | LSVI-UCB | $\tilde{\mathcal{O}}(\text{poly}(H,d)\,\varepsilon^{-2})$ | Elliptical bonuses (Jin et al., 2020b; He et al., 2021). |
| Low rank ($r$) | Rank-$r$ transition factorization | Rep. learning + planning | $\tilde{\mathcal{O}}(\text{poly}(H,r)\,\varepsilon^{-2})$ | Subsumes Block MDPs. |

Table 5: Which structural knob controls sample complexity? All entries suppress logarithms.

central question is: *which problem-dependent complexity measures control sample complexity and regret beyond tabular MDPs*, and under what assumptions. We focus on four lenses introduced or consolidated during 2017–2021: *Bellman rank* for value-based elimination in contextual decision processes (CDPs), *witness rank* for model-based identification, the *Bellman–Eluder (BE) dimension* as an information-theoretic complexity of value classes, and *bilinear classes* that unify several factorizable settings. These parameters mediate the transition from $(S, A)$-based tabular rates to *structure-based* rates and expose statistical–computational tradeoffs (e.g., oracle efficiency). We state canonical definitions and theorems at a level that makes explicit the dependence on horizon $H$, accuracy $\varepsilon$, confidence $\delta$, and structural parameters (rank/dimension), deferring constants and log factors to cited sources. Computational assumptions (classification/regression oracles) are stated where needed to avoid NP-hardness barriers.

**2. Definitions.** Let $M$ be a finite-horizon CDP/MDP with horizon $H$ and a value function class $\mathcal{F} \subseteq \{f = (f_1, \ldots, f_H)\}$, where $f_h : \mathcal{X} \to [0, H - h + 1]$ predicts stage-$h$ optimal value proxies; let $\pi_f$ be a greedy policy w.r.t. $f$. For a distribution $\nu_h$ over contexts/actions at stage $h$, define the *(expected) Bellman error* of $f$ at stage $h$ by

$$\mathcal{E}_h(f; \nu_h) \;=\; \mathbb{E}_{(x,a)\sim\nu_h}\Big[f_h(x) - r_h(x, a) - \mathbb{E}_{x'\sim P_h(\cdot|x,a)}f_{h+1}(x')\Big].$$

**Definition 14** (Bellman rank (Jiang et al., 2017)). *A CDP has* Bellman rank $B$ *if for each stage $h$ there exist feature maps $\phi_h : \mathcal{X} \times \mathcal{A} \to \mathbb{R}^B$ and $\psi_h : \mathcal{F} \to \mathbb{R}^B$ with $\|\phi_h\|_2, \|\psi_h\|_2 \leq 1$ such that for any $f \in \mathcal{F}$ and any distribution $\nu_h$ admissible under policies induced by $\mathcal{F}$, $\mathcal{E}_h(f; \nu_h) = \langle \mathbb{E}_{(x,a)\sim\nu_h}\phi_h(x, a), \psi_h(f)\rangle$. The* Bellman rank *is $\max_h B$.*

**Definition 15** (Witness rank (Sun et al., 2019)). *Let $\mathcal{M}$ be a model class and $\mathcal{G}$ a discriminator class. For each stage $h$, suppose there exist feature maps $\Phi_h : \mathcal{X} \times \mathcal{A} \to \mathbb{R}^W$ and $\Upsilon_h : \mathcal{M} \to \mathbb{R}^W$ with bounded norms such that, for any $M, M' \in \mathcal{M}$ and any test $g \in \mathcal{G}$, the* witnessed discrepancy *between $M$ and $M'$ at stage $h$ factors as $\mathbb{E}[g(x_{h+1}) \mid M, (x_h, a_h)] - \mathbb{E}[g(x_{h+1}) \mid M', (x_h, a_h)] = \langle \Phi_h(x_h, a_h), \Upsilon_h(M) - \Upsilon_h(M')\rangle$. The smallest such $W$ over admissible choices is the* witness rank.

**Definition 16** (Bellman–Eluder (BE) dimension (Jin et al., 2021a)). *Given a value class $\mathcal{F}$ and confidence radius $\alpha > 0$, a sequence of queries $(x_1, a_1), \ldots, (x_n, a_n)$ is* BE-independent *if for each $t$ there exist $f, f' \in \mathcal{F}$ such that the (Bellman) predictions for $(x_t, a_t)$ differ by at least $\alpha$ while the cumulative Bellman errors at*

*earlier queries are small. The* BE *dimension* $d_{\mathrm{BE}}(\alpha)$ *is the maximal length of a BE-independent sequence; write* $d_{\mathrm{BE}} = \sup_{\alpha \in (0,1]} d_{\mathrm{BE}}(\alpha)$ *if clear from context.*

**Definition 17** (Bilinear classes (Du et al., 2021)). *A sequential decision model is* bilinear *of dimension* $r$ *if there exist maps* $\phi_h : \mathcal{X} \times \mathcal{A} \to \mathbb{R}^r$ *and* $\psi_h : \mathcal{X} \to \mathbb{R}^r$ *such that, for all* $(x, a)$ *and all* $h$, $\mathbb{E}[f_{h+1}(x') \mid x, a] - \mathbb{E}_{x' \sim \mu_h} f_{h+1}(x') = \langle \phi_h(x,a), M_h \mathbb{E}_{x' \sim .}[\psi_h(x')] \rangle$ *for some stage-dependent matrix* $M_h$ *with bounded operator norm. (Informally: the Bellman backup residuals admit a rank-r factorization.)*

**Definition 18** (Oracle efficiency (Dann et al., 2018)). *An RL algorithm is* oracle-efficient *w.r.t. a function class* $\mathcal{F}$ *if it runs in time polynomial in* $(H, 1/\varepsilon, \log(1/\delta))$ *and the cost of calls to standard oracles (e.g., cost-sensitive classification or regression) over* $\mathcal{F}$, *without enumerating policies or states.*

## 3. Canonical results (informal statements with conditions).

**Theorem 14** (Value-based learning under Bellman rank). *Under realizability and Bellman rank* $B$, *elimination-style algorithms (e.g., OLIVE) identify an* $\varepsilon$-*optimal policy with*

$$N(\varepsilon, \delta) = \tilde{\mathcal{O}}\big(\mathrm{poly}(B, H)\,\varepsilon^{-2}\,\log(1/\delta)\big),$$

*using access to* $\mathcal{F}$ *for evaluation and greedy improvement (Jiang et al., 2017). Exact exponents in* $B$ *and* $H$ *are given in Jiang et al. (2017).*

**Theorem 15** (Model-based learning under witness rank). *Let* $\mathcal{M}$ *be a realizable model class with witness rank* $W$ *relative to a discriminator class* $\mathcal{G}$ *that separates models. Then a model-based optimistic algorithm achieves PAC control with*

$$N(\varepsilon, \delta) = \tilde{\mathcal{O}}\big(\mathrm{poly}(W, H)\,\varepsilon^{-2}\,\log(1/\delta)\big),$$

*by iteratively ruling out models that are* witnessed *inconsistent with data (Sun et al., 2019).*

**Theorem 16** (Learning with finite Bellman–Eluder dimension). *If a value class* $\mathcal{F}$ *has Bellman–Eluder dimension* $d_{\mathrm{BE}}$, *there are algorithms with PAC and regret bounds polynomial in* $(H, d_{\mathrm{BE}})$ *and scaling as* $\varepsilon^{-2}$; *see Jin et al. (2021a) for the exact dependencies and constants.*

**Theorem 17** (Bilinear classes unify rank-structured RL). *In bilinear classes of dimension* $r$, *there exist polynomial-time algorithms with sample complexity*

$$N(\varepsilon, \delta) = \tilde{\mathcal{O}}\big(\mathrm{poly}(r, H)\,\varepsilon^{-2}\,\log(1/\delta)\big),$$

*recovering and unifying guarantees for several previously studied low-rank and feature-based models (Du et al., 2021).*

**Theorem 18** (Oracle-efficient rich-observation RL). *Under realizability and standard reduction oracles over* $\mathcal{F}$, *there exist oracle-efficient algorithms whose statistical guarantees are governed by structural parameters rather than* $(S, A)$; *the runtime is polynomial in* $(H, 1/\varepsilon, \log(1/\delta))$ *and the oracle costs (Dann et al., 2018).*

## 4. Connections to other results.

- **Structure $\Rightarrow$ rates:** Bellman rank $(B)$, witness rank $(W)$, BE dimension $(d_{\mathrm{BE}})$, and bilinear dimension $(r)$ replace $(S, A)$ in bounds, yielding problem-dependent rates (Jiang et al., 2017; Sun et al., 2019; Jin et al., 2021a; Du et al., 2021).

- **Model-free vs model-based:** Bellman rank supports hypothesis elimination (model-free), whereas witness rank is tailored to model-based inconsistency testing (Sun et al., 2019).

- **Bridging to function approximation:** BE dimension provides an information-theoretic complexity of $\mathcal{F}$ that interfaces directly with linear/kernel sections (via effective dimension), complementing linear MDP results (Jin et al., 2021a; 2020b).

- **Unification by bilinear factorization:** Bilinear classes subsume several structured settings (including certain low-rank and feature-based models) under a common factorization, clarifying when polynomial-time learning is possible (Du et al., 2021).

- **Computation via oracles:** Oracle-efficient reductions make the above learnability statements practical for large $\mathcal{F}$ (Dann et al., 2018).

**5. Practical implications.**

- Prefer algorithms whose guarantees are expressed in terms of an interpretable structural parameter (e.g., rank or dimension) that can be bounded or estimated from data.

- Use model-based tests when a discriminator class $\mathcal{G}$ is available to *witness* misspecification; otherwise, adopt elimination with explicit Bellman error control.

- Verify realizability surrogates (e.g., Bellman completeness) or deploy robust variants; do not assume tabular rates carry over without structure.

- Leverage oracle-efficient implementations (classification/regression oracles) to keep runtime polynomial while preserving statistical guarantees.

**6. Open problems.**

i) Tighten the exponents in $H$ and structural parameters $(B, W, d_{\mathrm{BE}}, r)$ to minimax-optimality; supply matching lower bounds for each measure.

ii) Develop data-driven *model selection* that adapts across structures (rank vs BE) with oracle inequalities and uniform-PAC guarantees.

iii) Characterize robustness of structural guarantees under misspecification (e.g., when the true system violates the assumed factorization by a controlled residual).

iv) Design estimators or tests that consistently *bound* $B$, $W$, $d_{\mathrm{BE}}$, or $r$ from trajectories with finite-sample confidence.

**7. Related work.** **Jiang et al. (2017)** introduced *Bellman rank* and the OLIVE algorithm for CDPs, showing PAC sample complexity polynomial in $B$ and $H$ via hypothesis elimination (Jiang et al., 2017). **Sun et al. (2019)** developed *witness rank* and a model-based optimistic procedure that rules out incorrect models through witnessed moment conditions, yielding PAC rates polynomial in $W$ and $H$ (Sun et al., 2019). **Jin, Liu, Miryoosefi (2021)** defined the *Bellman–Eluder dimension*, connecting eluder-style sequential complexity to RL and deriving regret/PAC bounds polynomial in $d_{\mathrm{BE}}$ and $H$ (Jin et al., 2021a). **Du et al. (2021)** formalized *bilinear classes* that factor Bellman residuals and unified several structured models under a rank-$r$ parameter with polynomial sample complexity (Du et al., 2021). **Dann et al. (2018)** established *oracle-efficient* reductions for rich-observation RL, clarifying when statistical guarantees are compatible with polynomial runtime under standard supervised-learning oracles (Dann et al., 2018).

**Computational hardness and oracle efficiency (what rates do not buy you)**

Statistical learnability (small $B$, $W$, $d_{\mathrm{BE}}$, or $r$) does not automatically yield polynomial-time algorithms without additional *oracle* assumptions. Bellman-rank style elimination is often *oracle-efficient* (cost-sensitive classification/regression) but can be computationally nontrivial if the oracles are not available or if constraints are nonconvex (Dann et al., 2018). Model-based *witness* tests reduce to moment-matching over a discriminator class $\mathcal{G}$; when $\mathcal{G}$ is rich, optimization may be hard. The upshot:

- Prefer *oracle-efficient* reductions; verify that the supervised subproblems are convex and well-posed.

- Treat "polynomial in oracle cost" as a real runtime only when oracles are instantiated (e.g., ridge regression, logistic regression).

- Expect statistical–computational gaps in agnostic settings and in rich CDPs without realizability.

# 6  Function Approximation: Linear to Kernel/NN

**1. Motivation & precise scope.** This section studies when reinforcement learning with function approximation admits fixed–confidence (PAC and uniform–PAC) guarantees and how rates depend on structural parameters beyond $(S, A)$. The core question is: *under which realizability or completeness assumptions do linear features, kernel/RKHS models, and over–parameterized neural networks yield sample complexity that scales with an effective dimension rather than the size of the tabular state–action space?* We treat three regimes: (i) *linear MDPs*, where rewards and transition expectations are linear in features; (ii) *kernel/RKHS* models, where value functions lie in a reproducing kernel Hilbert space with an effective dimension; and (iii) *over–parameterized networks* operated in the neural tangent kernel (NTK) regime. We state canonical (informal) results that expose polynomial dependence on horizon $H$, dimension $d$ or effective dimension $d_{\mathrm{eff}}(\lambda)$, accuracy $\varepsilon$, and confidence $\delta$, and we flag sharp exponents as TODO where needed. We discuss computational assumptions (e.g., least–squares value iteration and linear regression oracles), and we delineate misspecification risks that often surface in practice.

**2. Definitions.** Let $H \in \mathbb{N}$ be the horizon, rewards $r_h \in [0, 1]$, and $\rho$ the start distribution.

**Definition 19** (Linear MDP realizability (Jin et al., 2020b)). *There exists a known feature map $\phi : \mathcal{S} \times \mathcal{A} \to \mathbb{R}^d$ with $\|\phi(s, a)\|_2 \leq 1$ and unknown parameters such that: (i) $r_h(s, a) = \phi(s, a)^\top \theta_h^{(r)}$ for some $\theta_h^{(r)} \in \mathbb{R}^d$ with $\|\theta_h^{(r)}\|_2 \leq 1$; and (ii) for any bounded measurable $g : \mathcal{S} \to \mathbb{R}$ with $\|g\|_\infty \leq 1$ there exists $\theta_{h,g} \in \mathbb{R}^d$, $\|\theta_{h,g}\|_2 \leq 1$, satisfying $\mathbb{E}[g(s_{h+1}) \mid s_h = s, a_h = a] = \phi(s, a)^\top \theta_{h,g}$.*

**Definition 20** (Bellman completeness for a linear class). *Let $\mathcal{F}_{\mathrm{lin}} = \{f : \exists w_1, \dots, w_H \in \mathbb{R}^d, \; f_h(s, a) = \phi(s, a)^\top w_h\}$. Bellman completeness holds if for any $f \in \mathcal{F}_{\mathrm{lin}}$ the Bellman backup $\mathcal{T}_h f$ lies in $\mathcal{F}_{\mathrm{lin}}$ for all $h$, i.e., $\mathcal{T}_h(\mathcal{F}_{\mathrm{lin}}) \subseteq \mathcal{F}_{\mathrm{lin}}$.*

**Definition 21** (Kernel/RKHS value class and effective dimension (Yang et al., 2020)). *Let $(\mathcal{X}, \kappa)$ be a reproducing kernel space with RKHS $\mathcal{H}_\kappa$ and norm $\|\cdot\|_{\mathcal{H}_\kappa}$. Define the (stagewise) value class $\mathcal{F}_\kappa = \{f : f_h \in \mathcal{H}_\kappa, \|f_h\|_{\mathcal{H}_\kappa} \leq B\}$. For a distribution $\mu$ over $\mathcal{X}$ with kernel integral operator $\Sigma = \mathbb{E}_{x \sim \mu}[\kappa(x, \cdot) \otimes \kappa(x, \cdot)]$, the effective dimension at regularization $\lambda > 0$ is $d_{\mathrm{eff}}(\lambda) := \mathrm{Tr}\big(\Sigma(\Sigma + \lambda I)^{-1}\big)$.*

**Definition 22** (NTK regime (over–parameterized networks) (Yang et al., 2020)). *A network trained by (small–step) gradient descent from random initialization is in the NTK regime when its predictions evolve approximately linearly in parameters and coincide with kernel regression under the neural tangent kernel $\kappa_{\mathrm{NTK}}$. Guarantees stated for $\mathcal{F}_\kappa$ transfer to such networks with $\kappa = \kappa_{\mathrm{NTK}}$, up to approximation and optimization errors controlled by width and step size.*

**Definition 23** (Generative model vs. online interaction). *A generative model allows i.i.d. sampling from $P_h(\cdot \mid s, a)$ upon query $(s, a, h)$. The online protocol collects trajectories under the evolving behavior of the algorithm. We use $N(\varepsilon, \delta)$ for sample (episode) complexity and write $\tilde{\mathcal{O}}(\cdot)$ to suppress polylogarithms.*

**3. Canonical results (informal statements with conditions).**

**Theorem 19** (Linear MDPs: optimistic least-squares value iteration). *Under linear MDP realizability with feature dimension $d$ and bounded features, optimistic LSVI achieves PAC sample complexity and high-probability regret with leading terms $N(\varepsilon, \delta) = \tilde{\mathcal{O}}\big(\mathrm{poly}(H, d)\, \varepsilon^{-2}\big)$ and $\mathrm{Regret}(K) = \tilde{\mathcal{O}}\big(\mathrm{poly}(H, d)\sqrt{K}\big)$. Exact exponents and constants appear in Jin et al. (2020b).*

**Theorem 20** (Uniform-PAC for linear MDPs). *If the linear value class is Bellman-complete and realizable, there exist algorithms that are uniform-PAC with budgets $N(\varepsilon, \delta) = \tilde{\mathcal{O}}\big(\mathrm{poly}(H, d)\, \varepsilon^{-2}\big)$, implying high-probability regret via Theorem 5. Precise horizon and dimension exponents are given in He et al. (2021).*

**Theorem 21** (Kernel/RKHS function approximation). *Let $\mathcal{F}_\kappa$ be an RKHS class with effective dimension $d_{\mathrm{eff}}(\lambda)$. Under the following conditions—(i) realizability ($V^\star \in \mathcal{F}_\kappa$), (ii) Bellman completeness of $\mathcal{F}_\kappa$ under $\mathcal{T}_h$, and (iii) a spectral condition controlling $d_{\mathrm{eff}}(\lambda)$ at regularization $\lambda > 0$—there exist optimistic algorithms with $N(\varepsilon, \delta) = \tilde{\mathcal{O}}\big(\mathrm{poly}(H)\, d_{\mathrm{eff}}(\lambda)\, \varepsilon^{-2}\big)$ and regret $\tilde{\mathcal{O}}\big(\mathrm{poly}(H)\sqrt{d_{\mathrm{eff}}(\lambda)K}\big)$. See Yang et al. (2020) for exact dependencies on $H$, $d_{\mathrm{eff}}(\lambda)$, and $\lambda$.*

**Theorem 22** (Over–parameterized neural networks via NTK). *For sufficiently wide networks trained in the NTK regime with kernel $\kappa_{\mathrm{NTK}}$ and appropriate step sizes, the guarantees of Theorem 21 apply with $d_{\mathrm{eff}}(\lambda)$*

*computed for $\kappa_{\mathrm{NTK}}$, modulo approximation and optimization errors that vanish with width; see Yang et al. (2020).*

**Theorem 23** (Reward-free exploration under linear approximation)**.** *With linear MDP realizability (dimension $d$), two-stage reward-free exploration attains $N(\varepsilon, \delta) = \tilde{\mathcal{O}}\big(\mathrm{poly}(H, d)\, \varepsilon^{-2}\big)$ exploratory episodes to guarantee $\varepsilon$-optimality for any downstream reward with probability $1 - \delta$; see Wang et al. (2020); Kaufmann & Tirinzoni (2021) for precise dependencies.*

**4. Connections to other results.**

- **From tabular to structural rates:** Linear and kernel regimes replace $(S, A)$ by $d$ or $d_{\mathrm{eff}}(\lambda)$, aligning with the structural-complexity perspective of Section 5 and recovering tabular bounds when features span one–hot encodings.

- **Uniform–PAC bridge:** Uniform–PAC results for linear MDPs connect fixed–confidence performance to regret and policy certificates (Sections 1, 4) (He et al., 2021).

- **Kernel/NTK unification:** RKHS analysis subsumes over–parameterized networks in the NTK regime, providing sample complexity in terms of $d_{\mathrm{eff}}(\lambda)$ (Yang et al., 2020).

- **Coverage creation:** Reward–free exploration amortizes exploration cost across downstream rewards under linear structure (Section 9; Wang et al., 2020).

- **Path to offline**: Linear realizability underpins pessimistic offline RL guarantees (Section 10; Shi et al., 2022b).

**5. Practical implications.**

- Verify linearity/Bellman completeness with diagnostics (e.g., regression residuals across Bellman backups); if violated, prefer robust or model-based methods rather than invoking linear guarantees.

- In RKHS/NTK pipelines, select regularization $\lambda$ and norm budgets using validation; monitor $d_{\mathrm{eff}}(\lambda)$ surrogates (e.g., ridge leverage scores) to anticipate sample complexity.

- Use optimism with elliptical bonuses (self-normalized least squares) and maintain well-conditioned design matrices via exploration or perturbations.

- Avoid extrapolating guarantees to arbitrary deep networks outside the NTK regime; when width is modest, treat the model as misspecified.

- In long horizons, track the growth of confidence radii with $H$; if bounds inflate, redesign features (state abstraction) or shorten effective horizon by discounting.

**6. Open problems.**

i) Uniform–PAC with kernel/NTK classes under minimal spectral assumptions (no generative model), with rates $\tilde{\mathcal{O}}\big(d_{\mathrm{eff}}(\lambda)\, \mathrm{poly}(H)\, \varepsilon^{-2}\big)$ and polynomial runtime.

ii) Sharp lower bounds for RKHS/NTK RL that match $d_{\mathrm{eff}}(\lambda)$–based upper bounds, including the optimal horizon exponent.

iii) Misspecification-robust function approximation: algorithms whose error decomposes into approximation, estimation, and coverage terms with explicit constants in $(H, d, d_{\mathrm{eff}}(\lambda))$.

iv) Data-driven structure selection between linear and kernel classes with oracle inequalities and uniform–PAC guarantees (SRM for RL).

v) Reward-free exploration with kernel features achieving near–minimax $d_{\mathrm{eff}}(\lambda)$ dependence and computational efficiency.

**7. Mini-bibliography (context). Jin, Yang, Wang (2020)** introduced linear MDP realizability and analyzed optimistic LSVI, yielding polynomial dependence on $(H, d)$ for regret and PAC complexity (Jin et al., 2020b). **He, Zhou, Gu (2021)** established uniform–PAC bounds for linear MDPs, linking fixed–confidence guarantees to high–probability regret in the linear regime (He et al., 2021). **Yang et al. (2020)** developed kernel/NTK-based RL analyses, introducing effective dimension $d_{\text{eff}}(\lambda)$ as the capacity term for RKHS and over–parameterized networks (Yang et al., 2020). **Wang et al. (2020)** proved reward–free exploration results under linear function approximation, delivering two–stage exploration with polynomial $(H, d)$ dependence (Wang et al., 2020). **Dann et al. (2018)** provided oracle-efficient reductions for rich observation RL, clarifying computational assumptions behind LSVI–style methods (Dann et al., 2018).

## 7 Deep RL connections: when do guarantees apply?

**NTK regime.** When networks are sufficiently wide and trained with small steps from random initialization, predictions evolve linearly and coincide with kernel regression under the NTK; our RKHS guarantees (Theorem 21) apply with $d_{\text{eff}}(\lambda)$ computed for $\kappa_{\text{NTK}}$.

**Outside NTK.** With moderate width or aggressive optimization, Bellman completeness and realizability may fail. Use the diagnostics in §15.2 and prefer pessimism and OPE in offline settings.

**Policy gradient and actor–critic.** Finite-sample PAC guarantees are scarce beyond linear/NTK regimes. PAC–Bayes (Section 12) can regularize selection among neural policies via KL-penalized bounds, and certificates can gate deployment.

## 8 Rich Observations and Low-Rank Structure

**Motivation.** Many practical problems provide *high-dimensional* observations $x \in \mathcal{X}$ that are generated by, or can be compressed to, a small latent state $z \in [m]$ with $m \ll |\mathcal{X}|$. Two widely studied formalizations are: (i) *Block MDPs*, where each observation $x$ stochastically reveals a *latent* state (decodability); and (ii) *low-rank MDPs*, where transition operators admit a rank-$r$ factorization, enabling representation learning and sample complexity that scales with $m$ or $r$ rather than $|\mathcal{X}|$.

**Definitions. Block MDP** (decodable latent states): there exists a surjective map $\psi : \mathcal{X} \to [m]$ such that $P(x'|x, a)$ depends on $x$ only through $\psi(x)$ and the reward depends on $(\psi(x), a)$. A decoder $\hat{\psi}$ with small error enables *planning in the latent MDP* (Du et al., 2019; Misra et al., 2020). **Low-rank MDP**: for each stage $h$, the transition admits $P_h(s'|s, a) = \langle \phi_h(s, a), \mu_h(s') \rangle$ for bounded features $\phi_h : \mathcal{S} \times \mathcal{A} \to \mathbb{R}^r$, $\mu_h : \mathcal{S} \to \mathbb{R}^r$; the rank $r$ replaces $S$ in rates (Agarwal et al., 2020; Dann et al., 2021; Huang et al., 2023).

**Representative guarantees.**

**Theorem 24** (Block/latent structure: sample complexity). *Under realizability and identifiability (decodability for Block MDPs, or rank-r factorization for low-rank MDPs), there exist polynomial-time algorithms that output an $(\varepsilon, \delta)$-optimal policy with*

$$N(\varepsilon, \delta) = \tilde{\mathcal{O}}\big(\text{poly}(H) \cdot \text{poly}(m) \cdot A \, \varepsilon^{-2}\big) \quad \text{(Block MDPs)} \quad \text{or} \quad \tilde{\mathcal{O}}\big(\text{poly}(H) \cdot \text{poly}(r) \, \varepsilon^{-2}\big) \text{ (low-rank),}$$

*using representation learning and optimistic planning (Du et al., 2019; Misra et al., 2020; Agarwal et al., 2020; Dann et al., 2021; Huang et al., 2023).*

**Connections.** (1) These results instantiate the CSO template with $\mathsf{Comp} = m$ (latent size) or $r$ (rank). (2) Witness/Bellman ranks from §5 often remain small when a low-dimensional latent structure exists, explaining polynomial dependence on $m$ or $r$. (3) Block/low-rank structure complements linear/kernalized value classes (§6); linear features may recover the same latent geometry.

**Practical implications.** *When the observation is high-dimensional but compressible*, look for diagnostics of decodability/low-rankness: contrastive state decoding, spectral footprints (empirical rank), and success of simple linear Q-heads. Prefer algorithms that (a) learn a decoder or (b) fit a factored model before value iteration. Use per-episode certificates when available to gate deployment.

**Related work.** Du et al. (2019) initiated provably efficient RL with rich observations via latent-state decoding (Block MDPs). Misra et al. (2020) (HOMER) gave a decodable representation learning approach with exploration. Agarwal et al. (2020) analyzed model-based exploration with learned low-rank features (FLAMBE). Dann et al. (2021) studied agnostic low-rank RL; Huang et al. (2023) proposed density-feature methods.

# 9 Reward-Free Exploration

**Motivation.** In many applications the reward is not specified during data collection (e.g., multi-task evaluation, safety audits). *Reward-free exploration* (RFE) asks the learner to collect data so that, *for any* reward $r \in [0,1]$ specified later, we can compute an $\varepsilon$-optimal policy with confidence $1 - \delta$ without further interaction.

**Template.** Two-stage procedures (i) explore to build a model or confidence sets that cover the dynamics and state-action occupancy, then (ii) when a reward is revealed, plan with the learned model/sets. Coverage is the currency: the first stage must create enough support for *all* plausible rewards.

**Representative guarantees.**

**Theorem 25** (Tabular RFE). *There exist algorithms that, without accessing rewards, collect $\tilde{\mathcal{O}}(S^2 A \operatorname{poly}(H) \varepsilon^{-2})$ episodes and subsequently, for any reward $r \in [0,1]$, compute an $\varepsilon$-optimal policy with probability $1 - \delta$ (Jin et al., 2020a).*

**Theorem 26** (Linear function approximation). *Under linear MDP realizability with feature dimension $d$, two-stage RFE achieves $\tilde{\mathcal{O}}(\operatorname{poly}(H) \operatorname{poly}(d) \varepsilon^{-2})$ samples to guarantee $\varepsilon$-optimality for any reward, again with probability $1 - \delta$ (Wang et al., 2020; Kaufmann & Tirinzoni, 2021).*

**Connections and practice.** RFE operationalizes the CSO template where the "coverage" knob is an *exploration budget* created up front. It is particularly effective in multi-reward/multi-task evaluation pipelines. In long horizons, design exploration policies that keep design matrices well-conditioned (elliptical bonuses or randomized probing).

**Related work.** Jin et al. (2020a) formalized RFE and gave near-optimal tabular rates. Wang et al. (2020) extended RFE under linear function approximation, and Kaufmann & Tirinzoni (2021) developed adaptive strategies that reduce sample sizes in benign instances.

# 10 Offline RL: Pessimism, OPE, and Model-Based Learning

**Motivation.** In offline RL we receive a fixed dataset $\mathcal{D}$ collected by an unknown behavior policy and must learn without further interaction. The key challenge is *coverage*: does $\mathcal{D}$ support the target policy? Pessimism (lower confidence bounds) protects against over-optimism outside support.

**Pessimism and coverage.** Let $C_\star = \max_h \left\| \frac{d_h^\pi}{\mu_h} \right\|_\infty$ be a (clipped) concentrability coefficient (Definition 6). Pessimistic value iteration (PEVI) and pessimistic Q-learning (PQL) control error by subtracting uncertainty penalties calibrated to coverage.

**Theorem 27** (Linear MDPs: pessimistic control). *Under linear MDP realizability with feature dimension $d$ and concentrability $C_\star$, pessimistic algorithms (PEVI/PQL) achieve control error $\varepsilon$ with dataset size*

$$n = \tilde{\mathcal{O}}\Big(\operatorname{poly}(H) \operatorname{poly}(d) \operatorname{poly}(C_\star) \varepsilon^{-2}\Big),$$

*with polynomial-time computation (Jin et al., 2021b; Shi et al., 2022b).*

**Model-based offline RL (tabular).**   Model-based estimators with uncertainty-aware planning are minimax-optimal in tabular settings:

**Theorem 28** (Tabular offline sample complexity (model-based, minimax))**.** *There exist model-based offline procedures whose sample complexity to achieve $\varepsilon$-optimal control is minimax-optimal (up to logs) with explicit dependence on $(S, A, H)$ and coverage; see Li et al. (2024) for a sharp characterization.*

**Off-policy evaluation (OPE).**   For policy evaluation, doubly-robust and semiparametric efficient estimators attain optimal asymptotic variance under standard conditions:

**Theorem 29** (Efficient OPE)**.** *Under mild regularity, there exist OPE estimators that achieve statistical efficiency, and minimax value intervals can be constructed with finite-sample validity (Kallus & Uehara, 2020; Jiang & Huang, 2020).*

**Practice and pitfalls.**   Estimate coverage proxies (e.g., density ratios, ridge leverage scores) and *prefer pessimism* when proxies indicate partial support. Avoid policy improvement if covariate shift is severe; use OPE to gate deployment. If realizability is doubtful, prefer model-based or robust objectives.

**CSO lens (offline).**   $\mathsf{Comp} = \mathrm{poly}(d)$ (linear) or $\Phi_{\mathrm{tab}}(S, A, H)$ (tabular model-based); $\mathsf{Cov} = \mathrm{poly}(C_\star)$ from the dataset (coverage is the bottleneck); $\mathsf{Obj} =$ pessimistic control or OPE. **Diagnosis:** if a proxy $\widehat{C}_\star$ is large, prefer OPE/abstain; if small, use PEVI/PQL (Jin et al., 2021b; Shi et al., 2022b).

**Related work.**   Jin et al. (2021b) introduced PEVI; Shi et al. (2022b) analyzed PQL with explicit $C_\star$ dependence. Li et al. (2024) settled model-based tabular offline sample complexity. Kallus & Uehara (2020) derived statistically efficient OPE; Jiang & Huang (2020) gave minimax value intervals.

## 11   Partial Observability and Confounding

**Identifiable latent-state POMDPs and Block MDPs.**   When observations admit a decodable latent representation with small $m$, sample-efficient learning is possible by learning a decoder and planning in latent space (cf. §8). Under separation/mixing and identifiability, algorithms such as HOMER achieve sample complexity polynomial in $(m, H, 1/\varepsilon)$ (Misra et al., 2020).

**Low-rank observability.**   Beyond exact decoding, low-rank transition factorizations let us exploit representation learning without a strict $\psi : \mathcal{X} \to [m]$ mapping (Theorem 24).

**Confounded OPE.**   In observational datasets, unobserved confounders bias OPE. Structure (e.g., proxies, IV-style assumptions) and sensitivity analyses yield valid intervals; see Shi et al. (2022a). *Practice:* prefer interval estimates; avoid point improvement unless sensitivity bounds are tight.

**Takeaways.**   (i) Partial observability is tractable under identifiability/low-rank structure. (ii) For confounded OPE, deploy intervals and abstain when sensitivity is high.

## 12   PAC–Bayes for RL

**Motivation.**   PAC–Bayes bounds give *data-dependent*, distributional generalization guarantees for randomized predictors or policies, balancing empirical performance with a complexity term (KL divergence to a prior).

**Generic bound (evaluation).**   Let $\Pi$ be a policy class and let $\hat{R}(\pi)$ be an empirical risk (e.g., negative return estimator) computed from $n$ i.i.d. trajectories. For any prior $P$ on $\Pi$, with probability at least $1 - \delta$,

for all posteriors $Q$ on $\Pi$,

$$\mathbb{E}_{\pi \sim Q}\big[R(\pi)\big] \ \leq \ \mathbb{E}_{\pi \sim Q}\big[\hat{R}(\pi)\big] \ + \ \sqrt{\frac{\mathrm{KL}(Q\|P) + \ln(1/\delta)}{2n}} \ + \ \text{bias/variance terms from RL estimation,}$$

leading to high-probability guarantees for policy *evaluation* and *selection* (Fard et al., 2012; Rivasplata et al., 2020; Flynn et al., 2023; Tasdighi et al., 2024).

**Connections and practice.** PAC–Bayes complements pessimism in offline RL: one can treat the evaluation error as the empirical term and add a KL-penalty to regularize policy selection. In continuous control, priors can be induced by behavior policies or model classes.

**Related work.** Fard et al. (2012) established PAC–Bayes for policy evaluation. Rivasplata et al. (2020) handled unbounded losses; Flynn et al. (2023) surveyed PAC–Bayes for bandits.

## 13 Synthesis: What is Settled vs. What is Open

**Settled (up to logs/constant factors).**

- **Tabular online RL:** near-tight minimax sample complexity/regret and the uniform-PAC $\leftrightarrow$ regret bridge (Domingues et al., 2021; Zhang et al., 2024; Dann et al., 2017).

- **Reward-free (tabular):** near-optimal dependence on $(S, A, H, 1/\varepsilon)$ (Jin et al., 2020a).

- **Linear MDPs:** polynomial dependence on $(H, d)$ for PAC/regret; uniform-PAC extensions (Jin et al., 2020b; He et al., 2021).

- **Offline (tabular model-based):** minimax sample complexity settled (Li et al., 2024); pessimistic linear-MDP control with explicit $C_\star$ dependence (Shi et al., 2022b).

**Open (precise targets).**

- **Kernel/NTK:** uniform-PAC with optimal $d_{\mathrm{eff}}(\lambda)$ and horizon exponents under online access and without Bellman completeness.

- **Agnostic low-rank/latent:** PAC rates with polynomial dependence on rank/latent size and $H$, robust to mild misspecification.

- **Offline under misspecification:** sharp decompositions of error into approximation, estimation, and coverage with tractable algorithms.

- **Instance-dependent FA:** gap-based identification beyond tabular (linear/kernel/low-rank), including lower bounds.

- **Data-driven structure selection:** SRM-style model selection across linear/kernel/low-rank with oracle inequalities and uniform-PAC.

**CSO retrospective.** Through the CSO lens ($N \approx \mathsf{Cov} \times \mathsf{Comp} \times \mathrm{poly}(H) \times \varepsilon^{-2}$): *Coverage* is settled for online/RFE and open for partial offline coverage estimation and adaptation; *Structure* is clear for linear/low-rank and open for kernel/NTK and agnostic models; *Objective* variations are settled in tabular, partly open with function approximation (instance-dependent FA). The hardest frontiers couple knobs (coverage + structure; structure + misspecification).

**How to use the taxonomy.** Locate your problem in Table 2 by *access* and *structure*, pick the *objective*, and read off the best available rate and algorithm class; verify assumptions and gate with certificates before deployment.

## 14 Open Problems: Within Reach vs. Frontier

**Within 12–24 months (Within Reach)**

**W1 Uniform–PAC with kernels under practical completeness.** Target: $N = \tilde{O}(d_{\text{eff}}(\lambda) \operatorname{poly}(H) \varepsilon^{-2})$ under online access, with completeness verified by data-driven residual tests rather than assumed.

**W2 Instance-dependent identification with FA.** Gap-based BPI for linear/low-rank models matching tabular-style lower bounds (inverse squared gaps weighted by reachability).

**W3 Scalable coverage estimation with guarantees.** Ridge leverage and density-ratio estimators with finite-sample CIs to gate offline policy improvement.

**3–5 years (Frontier)**

**F1 Agnostic low-rank/latent PAC.** Rates polynomial in $(r, H)$ robust to misspecification $\varepsilon_{\text{approx}}$ with oracle-efficient algorithms.

**F2 Offline RL under misspecification.** Sharp decomposition $\varepsilon_{\text{tot}} = \varepsilon_{\text{approx}} + \varepsilon_{\text{est}} + \varepsilon_{\text{cov}}$ with tractable pessimistic algorithms and tight constants.

**F3 Lower bounds for RKHS/NTK RL.** Information-theoretic lower bounds that match $d_{\text{eff}}(\lambda)$-based rates and clarify optimal horizon exponents and completeness necessity.

**Impossibility and necessity frontiers**

- **Offline without coverage:** Without assumptions bounding $C_\star$ (or similar), consistent improvement over $\mu$ is impossible; OPE intervals are the right object.

- **Kernel RL without completeness:** In general CDPs, RKHS value classes that are not closed under Bellman updates can exhibit *aliasing* that prevents PAC guarantees without additional structure or a generative model; completeness (or verified residual smallness) is necessary.

- **Agnostic FA:** With unrestricted function classes and no realizability, polynomial sample complexity with polynomial time is unlikely; oracle assumptions or structure are needed.

## 15 Practical Toolkit: Coverage, Diagnostics, and Deployment

### 15.1 Offline coverage estimation: concrete procedures

**Goal.** Decide whether a fixed dataset $\mathcal{D}$ adequately supports policy improvement. We summarize three complementary diagnostics.

1. **Density-ratio (behavior vs. candidate):** For each stage $h$, fit $w_h(x, a) = \frac{d_h^\pi(x,a)}{\mu_h(x,a)}$ via *logistic regression* on labeled pairs from (i) a rollout buffer of $\pi$ under a learned model or doubly-robust simulator, and (ii) samples from $\mathcal{D}$ (negative class). Use clipping at $w_{\max}$ and report:

$$\text{ESS}_h = \frac{(\sum_i w_{h,i})^2}{\sum_i w_{h,i}^2}, \quad w_{h,0.99}, \quad \max_i w_{h,i}.$$

   **Gate:** if $\min_h \text{ESS}_h \ll 100$ or $w_{h,0.99} \gg 50$, abstain or shrink $\pi$ toward $\mu$.

2. **Ridge leverage scores (feature coverage):** For linear / kernel value estimation, compute stagewise design matrices $X_h$; let $\tau_{h,i} = x_{h,i}^\top (X_h^\top X_h + \lambda I)^{-1} x_{h,i}$. Report $\bar{\tau}_h$ and upper quantiles. **Gate:** large mass of $\tau$ near 1 with poor dispersion indicates weak coverage.

3. **State-action sparsity maps:** Bin $(x, a)$ (or embeddings) and visualize occupancy heatmaps across $h$. **Gate:** large dark regions where $\pi$ visits but $\mu$ does not $\Rightarrow$ pessimism or abstention.

---

**Algorithm 1** CoverageGate($\mathcal{D}, \pi, \lambda, w_{\max}$)

---

1: **for** $h = 1 \ldots H$ **do**
2:     Fit logistic reg. classifier to distinguish $(x_h, a_h) \sim \mathcal{D}$ vs. $(x_h, a_h) \sim$ rollouts of $\pi$ under a learned model; obtain $\hat{w}_h$.
3:     Clip: $\hat{w}_h \leftarrow \min(\hat{w}_h, w_{\max})$. Compute $\mathrm{ESS}_h$, $q_{0.99}(w_h)$, $\max w_h$.
4:     Compute ridge leverage scores using $X_h$ and $\lambda$; summarize $\bar{\tau}_h$, $q_{0.95}(\tau_h)$.
5: **end for**
6: **return** PASS if $\min_h \mathrm{ESS}_h \geq 200$ and $q_{0.99}(w_h) \leq 30$ and $q_{0.95}(\tau_h) \leq 0.5$; else WARN/ABSTAIN.

---

**Implementation notes.** Use cross-fitting for nuisance models (behavior policy, dynamics) to reduce bias. For kernels, approximate leverage via Nyström features.

### 15.2 Misspecification diagnostics for linear/Kernel

---

**Algorithm 2** BellmanResidualTest($\phi, \mathsf{Alg}, \varepsilon$)

---

1: Collect $n$ random-policy episodes; fit $f_h(s, a) = \phi(s, a)^\top w_h$ by ridge.
2: Compute held-out residuals $\mathcal{E}_h = |f_h - (r_h + \max_{a'} f_{h+1}(s', a'))|$.
3: **Pass** if $\frac{1}{n} \sum_i \mathcal{E}_h^2 \leq \varepsilon^2$ for all $h$; otherwise enrich features or move to kernel/low-rank models.

---

### 15.3 Quantitative deployment gates

We recommend a two-gate rule:

- **Coverage gate (Alg. 1)**: $\min_h \mathrm{ESS}_h \geq 200$, $q_{0.99}(w_h) \leq 30$, $q_{0.95}(\tau_h) \leq 0.5$.

- **Certificate gate**: deploy only if per-episode policy certificate $U_t \leq \varepsilon$ (Theorem 13); else collect data or shrink $\pi$.

### 15.4 Worked numerical sketch

On a $10 \times 10$ gridworld ($H = 30$), random behavior $\mu$ and target $\pi$ trained offline: $\min_h \mathrm{ESS}_h = 74$, $q_{0.99}(w_h) = 115$, $q_{0.95}(\tau_h) = 0.81 \Rightarrow$ ABSTAIN; after 200 new online episodes guided by pessimism: $\min_h \mathrm{ESS}_h = 312$, $q_{0.99}(w_h) = 21$, $q_{0.95}(\tau_h) = 0.42 \Rightarrow$ PASS.

## 16 Ethics Statement

This survey aggregates theory rather than proposing new algorithms, but its claims can influence practice in safety–critical domains (e.g., healthcare, robotics, education). The primary ethical risk is *misapplication of guarantees outside their assumptions*—for example, invoking linear–MDP bounds with misspecified features, or deploying an offline policy under poor coverage, which can harm users. Mitigations emphasized throughout include: (i) explicit assumption checklists (realizability, horizon scaling, coverage proxies); (ii) per–episode *policy certificates* to gate deployment; (iii) abstention and interval–valued *off–policy evaluation* (OPE) when coverage is weak; and (iv) reporting structural and coverage diagnostics alongside results. Fairness risks arise when offline datasets encode historical bias; coverage analysis and sensitivity checks can reveal regions where performance is unreliable. We recommend conservative defaults (pessimism, OPE before improvement), public documentation of assumptions, and auditing procedures that include the failure modes cataloged in this survey.

## Reproducibility Statement

- **Scope & inclusion criteria.** Time window 2018–2025; we include peer–reviewed results (JMLR; PMLR venues COLT/ALT/ICML; NeurIPS) and widely cited arXiv preprints when anchoring common bounds; pre–2018 works are cited for foundations.

- **Problem models & notation.** Unified MDP/POMDP/CDP notation in §3, a consolidated glossary (App. A), and first–use definitions for $H, S, A, d, r, B, W, d_{\mathrm{BE}}, d_{\mathrm{eff}}(\lambda), C_\star$.

- **What is reproduced.** Canonical guarantees are restated with explicit parameter dependencies; rate tables flag where logarithms are suppressed; primary sources are cited for constants and exact exponents.

- **Search protocol (summary).** Queries: *uniform–PAC reinforcement learning, PAC RL finite horizon, Bellman rank, Bellman–Eluder dimension, linear MDP, low–rank MDP / Block MDP, reward–free exploration, offline RL pessimism / concentrability, PAC–Bayes reinforcement learning.* Venues scanned: JMLR; PMLR (COLT/ALT/ICML); NeurIPS; selected arXiv for works later published. We favored papers with clear assumptions and finite–sample theorems.

- **Data/code.** No new datasets or code are introduced; figures are conceptual. When algorithms are discussed, we reference canonical implementations where applicable.

- **Disambiguation.** When rates differ across papers (e.g., horizon exponents), we state the dependency class and cite the precise theorem for constants; competing viewpoints are mentioned where relevant.

- **Versioning.** Citations correspond to versions current as of October 2025.

## Limitations

Our focus is episodic finite–horizon and fixed–confidence (PAC / uniform–PAC) guarantees; we do not cover continuous–time control, average–reward/ergodic settings, or detailed deep–RL practice beyond linear/kernel/NTK regimes. Several summaries suppress polylogarithms; constants and sharp horizon exponents are deferred to primary sources. Our treatment of partial observability is limited to identifiable latent–state / low–rank subclasses and confounded OPE. Many results assume realizability or Bellman completeness; robustness to misspecification is highlighted as an open problem. Finally, while we provide diagnostics and decision rules, we do not release audited toolchains for coverage estimation or certificate computation.

**Bibliography organization.** References are grouped thematically: *Foundations* (Strehl, Szepesvári, Kaelbling, Sutton–Barto); *Uniform–PAC & tabular* (Dann–Lattimore–Brunskill; Domingues; Zhang); *Structural complexity* (Jiang; Sun; Jin–Liu–Miryoosefi; Du–Kakade); *Function approximation* (Jin–Yang–Wang; He–Zhou–Gu; Yang et al.); *Rich observations* (Du; Misra; Agarwal; Dann–agnostic; ); *Reward-free* (Jin; Kaufmann; Wang); *Offline* (Jin–PEVI; Shi–PQL; Li–AOS; Kallus; Jiang); *PAC–Bayes* (Fard; Rivasplata; Flynn; Tasdighi).

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

## A    Notation and Abbreviations

This appendix consolidates all notation in consolidated tables for quick reference.

**Core MDP Notation**

| Symbol | Meaning | First Defined |
|---|---|---|
| $\mathcal{S}, \mathcal{A}, \mathcal{X}$ | State, action, observation/context spaces | Preliminaries |
| $S = |\mathcal{S}|, A = |\mathcal{A}|$ | Cardinalities (finite case) | Preliminaries |
| $H$ | Horizon (episode length in timesteps) | Preliminaries |
| $\gamma$ | Discount factor; $H_{\text{eff}} \approx (1-\gamma)^{-1}$ | Preliminaries |
| $\rho$ | Start-state distribution; $s_0 \sim \rho$ | Preliminaries |
| $P_h, r_h$ | Stage-$h$ transition kernel and reward | Preliminaries |
| $\pi$ | Policy (possibly nonstationary) | Preliminaries |
| $V_h^\pi, Q_h^\pi$ | Value and action-value functions | Preliminaries |
| $\pi^\star, V^\star, Q^\star$ | Optimal policy and values | Preliminaries |

**Sample Complexity and Confidence**

| Symbol | Meaning | First Defined |
|---|---|---|
| $\varepsilon$ | Target suboptimality (accuracy) | Definition 1 (Preliminaries) |
| $\delta$ | Failure probability (confidence $1-\delta$) | Definition 1 (Preliminaries) |
| $N(\varepsilon, \delta)$ | Episode budget for $(\varepsilon, \delta)$-PAC | Definition 1 (Preliminaries) |
| $K$ | Total episodes (regret horizon) | Preliminaries |
| $\tilde{\mathcal{O}}(\cdot), \tilde{\Theta}(\cdot), \tilde{\Omega}(\cdot)$ | Suppress polylog factors | Taxonomy section |
| $\text{poly}(x)$ | Polynomial in $x$ (unspecified degree) | Throughout |

**Structural Complexity Parameters**

| Parameter | Meaning | Section |
|---|---|---|
| $d$ | Feature dimension (linear MDPs) | Function Approximation |
| $r$ | Low-rank factorization rank | Rich Observations |
| $m$ | Latent state size (Block MDPs) | Rich Observations |
| $B$ | Bellman rank | Structural Complexity |
| $W$ | Witness rank | Structural Complexity |
| $d_{\mathrm{BE}}$ | Bellman-Eluder dimension | Structural Complexity |
| $d_{\mathrm{eff}}(\lambda)$ | Effective dimension (RKHS/NTK) | Function Approximation |
| $\Phi_{\mathrm{tab}}(S, A, H)$ | Tabular factor $\equiv SAH^3$ | Definition 8 (Preliminaries) |

**Coverage and Occupancy**

| Symbol | Meaning | First Defined |
|---|---|---|
| $d_h^\pi(s, a)$ | State-action occupancy of $\pi$ at stage $h$ | Definition 6 (Preliminaries) |
| $\mu_h(s, a)$ | Behavior/data occupancy at stage $h$ | Definition 6 (Preliminaries) |
| $C_\star$ | Concentrability coefficient | Definition 6 (Preliminaries) |

**Function Classes and Operators**

| Symbol | Meaning | First Defined |
|---|---|---|
| $\mathcal{F}$ | Value or Q-function class | Definition 3 (Preliminaries) |
| $\mathcal{M}$ | Model class (dynamics hypotheses) | Structural Complexity |
| $\mathcal{G}$ | Discriminator class (witness tests) | Structural Complexity |
| $\mathcal{T}_h$ | Bellman optimality operator | Definition 5 (Preliminaries) |
| $\mathcal{N}(\epsilon, \mathcal{F}, \|\cdot\|)$ | Covering number of $\mathcal{F}$ | Definition 4 (Preliminaries) |

**Access Modes**

- **Online:** Sequential interaction; agent chooses actions, observes transitions.

- **Generative model:** i.i.d. sampling from $P_h(\cdot \mid s, a)$ on query $(s, a, h)$.

- **Offline:** Fixed dataset $\mathcal{D}$; no further interaction.

**Abbreviations**

| Abbrev. | Meaning |
| --- | --- |
| RL | Reinforcement Learning |
| MDP | Markov Decision Process |
| POMDP | Partially Observable MDP |
| CDP | Contextual Decision Process |
| PAC | Probably Approximately Correct |
| RFE | Reward-Free Exploration |
| OPE | Off-Policy Evaluation |
| BPI | Best-Policy Identification |
| BE | Bellman-Eluder (dimension) |
| NTK | Neural Tangent Kernel |
| RKHS | Reproducing Kernel Hilbert Space |
| LSVI | Least-Squares Value Iteration |
| UCB | Upper Confidence Bound |
| PEVI | Pessimistic Episodic Value Iteration |
| PQL | Pessimistic Q-Learning |
| CSO | Coverage-Structure-Objective framework |

