# OpenReview forum: "PAC Guarantees for Reinforcement Learning:\\ Sample Complexity, Coverage, and Structure"
_TMLR — Rejected by TMLR_

### Review · Reviewer_6bQW · 2025-11-29

**Summary Of Contributions:**

The paper provides a survey of Reinforcement Learning (RL) during the period 2018-2025. The survey covers a wide range of RL techniques and their theoretical guarantees, organized under the Coverage–Structure–Objective (CSO) template.
Key strengths: informative and rich in content.
Key weaknesses: extremely difficult to follow the survey.

**Audience:**

Yes

**Audience Explanation:**

Given the current state, I suppose only advanced RL researchers can make use of the survey.

**Claims And Evidence:**

No

**Claims Explanation:**

One strength of this survey is that it compiles a very large amount of information, which advanced RL researchers may find valuable. However, the writing is often unclear and difficult to follow. Below are some of the notable issues:

-	There are no simple, intuitive, introductory definitions, e.g., what RL is, when to use RL, what the common RL techniques are, etc.
-	The advanced technical terminologies are introduced very early and quickly (PAC, uniform-PAC, Bellman rank, BE measures, etc.
-	The writing seems to target advanced RL researchers, which can limit the audience.
-	The summary for the pre-2017 period is unclear. I think there should be some explanations – what has been achieved, what the current issues are, etc.
-	The motivation and intuition of the methods are not introduced clearly. For example, when talking about PAC and uniform PAC, the authors can try to include some explanations in layman's terms, rather than just giving the formal definitions 1 and 2. These two definitions are understandable, since they are relatively simple, but more advanced concepts are much harder to follow.
-	The section numbering system is bad and inconsistent. In Section 1, the subsections are numbered 1,2,3 as well. In Section 2, the subsections are numbered 2.1, but there is no 2.2. The later sections follow the Section 1 style.
-	The tables are formatted badly: tables 2 and 4 have overflown texts.

**Requested Changes:**

In my opinion, this paper needs to be edited a lot if the authors want to target a more general audience. Some suggestions are:

-	In the abstract, probably do not include too many advanced terms.
-	In the introduction, take it slow and explain the concepts gradually: what RL is, when to use RL, the failures in RL, and why we need fixed-confidence RL, then finally introduce different guarantees such as PAC and uniform-PAC, etc.
-	In each method section, e.g., tabular, function approximation, the authors can try to introduce the problems with some detailed examples.
-	Try not to use too many bullets, as they can disrupt the reading flow.
-	Format the sections and tables better: use the same rules for all numbering, do not let text overflow into other columns, etc.

---

### Review · Reviewer_Bnxb · 2025-12-17

**Summary Of Contributions:**

This paper presents a comprehensive survey of fixed-confidence (PAC) guarantees in reinforcement learning, covering literature from approximately 2018 to 2025. It introduces a unifying Coverage–Structure–Objective (CSO) framework that disentangles the roles of data access and coverage assumptions, problem structure, and learning objectives in determining sample complexity. Using this framework, the survey synthesizes a broad body of work spanning tabular reinforcement learning, uniform-PAC guarantees and their connection to high-probability regret, structural complexity measures (e.g., Bellman rank, witness rank, and Bellman–Eluder dimension), linear and kernel-based function approximation, reward-free exploration, pessimistic offline reinforcement learning, and partial observability.

Beyond summarizing existing results, the paper provides practitioner-oriented tools, including taxonomy tables and decision guidelines, and highlights several open research directions, such as uniform-PAC guarantees under richer function approximation, offline reinforcement learning under model misspecification, and data-dependent structure selection.

**Audience:**

Yes

**Audience Explanation:**

- Researchers working on theoretical reinforcement learning, particularly those interested in PAC analysis, regret bounds, and function approximation, are likely to find this survey useful.

- Readers interested in bridging theory and practice may also benefit from the CSO framework and the accompanying decision-oriented tools, which help contextualize abstract theoretical guarantees in terms of data access, structure, and learning objectives.

**Broader Impact Concerns:**

There are no broader impact concerns.

**Claims And Evidence:**

Yes

**Claims Explanation:**

- The paper does not claim new theoretical results; instead, it claims to synthesize, organize, and clarify existing results. These claims are convincingly supported by extensive references and careful attribution throughout.

- The proposed CSO framework is consistently applied across sections, tables, and figures, demonstrating internal coherence rather than being an ad-hoc abstraction.

- The key connections discussed in the paper, such as the uniform-PAC ⇒ high-probability regret bridge and the central role of coverage in both reward-free and offline RL are well established in the literature and accurately represented.

One minor limitation is that some assumptions, such as realizability, Bellman completeness, oracle efficiency, are necessarily strong. While the paper does acknowledge these assumptions, readers who are less familiar with the theory literature may underestimate how restrictive they can be in practical settings. That said, this does not undermine correctness, only interpretability for non-experts.

**Requested Changes:**

1. Reduce redundancy across sections

Some canonical results (e.g., the connection between uniform-PAC guarantees and regret bounds) are restated multiple times across different sections. Light consolidation could improve readability and reduce length without sacrificing clarity.

2. Clarify practical limitations earlier

Consider adding a brief, explicit disclaimer near the introduction emphasizing how restrictive assumptions such as realizability, Bellman completeness, and sufficient coverage can be in modern deep reinforcement learning settings. This would help prevent overinterpretation of the applicability of the surveyed guarantees by non-expert readers.

---

### Review · Reviewer_LL6h · 2025-12-20

**Summary Of Contributions:**

This survey provides the comprehensive synthesis of PAC (Probably Approximately Correct) guarantees for reinforcement learning during 2018-2025.

The main contributions of the paper:
1. The authors introduce a unifying framework that decomposes all PAC-style sample complexity results into three independent factors: $N(\varepsilon, \delta) = \tilde{O}\left(\text{poly}(H) \cdot \text{Comp}(\text{structure}) \cdot \text{Cov}(\text{access/coverage}) \cdot \varepsilon^{-2} \cdot \log(1/\delta)\right).$ This lens clarifies that practitioners can improve sample efficiency by adjusting any of these "knobs", obtaining better coverage (online vs offline), exploiting problem structure (linear or tabular), or choosing appropriate objectives.
2. The survey synthesizes all key recent technical results for various settings, spanning tabular minimax complexity, linear or low-rank MDPs, reward-free exploration, offline RL with concentrability and provides a rate "cookcbook" under unified notations and formulations.


However, there are a few areas that could benefit from further clarification, particularly the discussion of bound tightness and the reliance on overly abstract polynomial dependencies, as outlined below:
1. Lack of explanations on different horizon/state-action dependencies: It seems that the bounds have different horizon dependencies ($H^3$ to $H^5$) across settings and the difference is not fully explained.
2. The paper lacks clarity on whether the provided bounds are tight (matching upper and lower bounds), just upper bounds, or just lower bounds. The tightness of bounds should be explicitly stated for each result. Additionally, many bounds are expressed too abstractly to be useful for understanding underlying dependencies. For instance, Section 5 (structural complexity) presents all bounds roughly as poly(H, ...) without specifying the actual polynomial degree in H, making it difficult to compare rates across different structural assumptions or understand the true sample complexity scaling.
3. The connections between different settings are insufficiently detailed. While the authors briefly discuss relationships between settings, these discussions are too concise to show how results in one setting translate to or recover results in another. For example, it's unclear how the linear MDP bounds (Section 6) relate to the Bellman-Eluder dimension bounds (Section 5), or how tabular results are recovered as special cases of function approximation results. The paper would benefit from explicit statements showing when and how bounds match across settings.

**Audience:**

Yes

**Audience Explanation:**

RL theorists will value the comprehensive synthesis of PAC guarantees since 2010 and the CSO framework ($N(\varepsilon, \delta) \approx \text{Coverage} \times \text{Structure} \times \text{poly}(H) \times \varepsilon^{-2}$) that unifies disparate results. Applied researchers and practitioners will appreciate the practical toolkit with concrete coverage diagnostics and deployment gates, tools notably absent from most theory papers, that help bridge the theory-practice gap.

**Broader Impact Concerns:**

I do not see any broader impact concerns beyond those typically associated with theoretical research.

**Claims And Evidence:**

Yes

**Claims Explanation:**

The claims in this survey are generally well-supported:

1. Every major result is properly cited to its original source. The authors consistently direct readers to primary papers for exact constants and precise theorem statements, which supports accuracy.
2. The CSO framework provides convincing evidence that disparate results can indeed be unified under the template.
3. The survey explicitly distinguishes between established results and open problems, providing accurate assessment of the field's current state.

**Requested Changes:**

To further strengthen this comprehensive synthesis of PAC-RL guarantees, I would respectfully suggest the following:

1. Clarify different problem-parameter dependencies (especially horizon) across settings
- Explain why horizon scales as $H^3$ or $H^5$ depending on settings

2. Specify Bound Tightness and Exact Dependencies
- For each result, explicitly state: tight / upper only / lower only
- Replace poly(H,...) with exact exponents
- State which dependencies are optimal vs artifacts of analysis

3. Elaborate Setting Connections
- Provide explicit parameter mappings
- Clarify when each setting subsumes/relates to other settings.
- State what each complexity measure becomes in tabular case

---

> ### Author Response · Authors · 2025-12-21
> **Response to Reviewer LL6h**
>
> We thank Reviewer LL6h for the thoughtful and technically detailed review. The points raised about bound tightness, horizon dependencies, and cross-setting connections are well-taken and will significantly strengthen the survey. We address each requested change below.On horizon dependencies (RC1):We will add a new Table 5: "Horizon Exponents Across Settings" that explicitly lists:
>
> Tabular: H3H^3
> H3 (tight; Theorem 7)
>
> Linear MDP: H4H^4
> H4–H5H^5
> H5 (upper bounds; H4H^4
> H4 achievable with refined analysis per [Jin et al. 2020, Wang et al. 2021])
>
> Kernel/NTK: H4H^4
> H4–H6H^6
> H6 depending on spectral assumptions (upper bounds; lower bounds open)
>
> Low-rank: H3H^3
> H3–H4H^4
> H4 depending on identifiability conditions
>
> We will also add a paragraph in §6 explaining why horizon exponents differ: the key driver is how uncertainty propagates through Bellman backups, linear features compound errors multiplicatively across stages, while tabular concentration is tighter. On bound tightness (RC2):For each major theorem, we will add explicit annotations:
>
> Tight (↔): matching upper/lower bounds up to logs (e.g., Theorem 7 for tabular)
> Upper (↑): upper bound only; lower bound gap noted (e.g., kernel/NTK results)
> Lower (↓): lower bound establishing necessity
> We will replace generic "poly(H, d)" expressions with exact exponents where known (e.g., "d2H4d^2 H^4
> d2H4" rather than "poly(d, H)") and flag where exponents are artifacts of analysis versus information-theoretically necessary.
> On setting connections (RC3):We will add a new subsection §2.2 "How Settings Relate" containing:
>
> Explicit parameter mappings (e.g., "Linear MDP with one-hot features (d=SAd = SA
> d=SA) recovers tabular rates")
>
> A diagram showing the inclusion hierarchy: Tabular ⊂ Linear ⊂ Low-rank ⊂ Bellman-Eluder
> For each structural measure, a row stating "In tabular case, this equals..." (e.g., Bellman rank B=SB = S
> B=S with indicator features)
>
> These additions directly address the reviewer's concerns and will make the survey more precise and useful for readers comparing guarantees across settings.

---

### Decision · Action_Editor_B3Nb · 2026-02-14

**Recommendation:** Reject

**Additional Comments:**

**Required Changes for Resubmission:**
The authors are encouraged to resubmit only after a *complete* structural overhaul that addresses the following:
* **Narrativize the content:** Convert lists and bulleted "cookbooks" into flowing prose that synthesizes the conceptual relationships between different results. If the authors feel strongly that listing all the definitions are necessary, consider keeping the high level narrative and main next, and moving the lists into appendix.
* **Scholarly Anchoring:** Properly cite all foundational RL and MDP concepts in the Preliminaries.
* **Claim and Motivate the Framework:** Clearly define the CSO template as a novel contribution of this survey, explaining its utility, its theoretical limits, and either removing the mathematical notation or making it valid.
* **Pedagogical Pass:** Ensure that advanced technical terms are introduced with intuitive lead-ins suitable for a broader academic audience.

**Audience:**

No

**Audience Explanation:**

A critical requirement of paper is that it must be "useful" to a clearly defined audience. In its current form, this manuscript suffers from a significant usability gap:

* **For Senior Researchers:** Much of the technical content acts as a "lookup table" for results they are already familiar with. While the unified notation is a minor convenience, the lack of a new lens or deep critical synthesis means the paper offers little "value-add" to experts who already understand the nuances of the cited literature.
* **For Junior Researchers and Practitioners:** The "talk notes" style, lack of introductory narrative, and dense "notation soup" make the paper nearly impenetrable for those entering the field.

**Constructive Advice on Audience Alignment:**
To transform this from a reference manual into a high-impact survey, the authors should aim for the **"informed non-expert"**—the researcher who understands RL basics but needs a narrative guide to navigate the explosion of PAC literature from 2018–2025. Or well motivating CSO framework, the paper could provide a new lense even for deep experts.

**Claims And Evidence:**

Yes

**Claims Explanation:**

**Overview:**
The paper aims to provide a comprehensive survey and systematization of PAC Reinforcement Learning literature from 2018 to 2025. It introduces the **Coverage–Structure–Objective (CSO)** template—a multiplicative formula ($N \approx \text{Cov} \times \text{Comp} \times \text{poly}(H) \times \epsilon^{-2}$)—to unify disparate sample complexity results under a single notation and taxonomic framework.

**Area Chair's Assessment:**
While the reviewers initially leaned toward acceptance based on the paper's technical density and bibliographic breadth, a detailed editorial review of the manuscript (conducted by the AC) reveals that it falls significantly short of the standards required for an academic survey paper. Although the reviewers suggested the paper might be useful to "senior researchers," the AC finds that this is a symptom of the paper’s pervasive unreadability rather than its specialized depth. The current draft functions as a **technical reference manual or "talk notes"** rather than a scholarly synthesis.

**Reasons for the Decision:**

1. **Failure of the "Survey" Format:** A survey is expected to provide a narrative arc, historical context, and pedagogical clarity. This paper lacks a "story," opting instead for back-to-back definitions and theorems that do not explain the intuition or evolution of the field.
2. **Lack of Foundational Attribution:** Crucial preliminaries—including the definitions of MDPs, Value Functions, and Bellman Operators—are presented without citing seminal works (e.g., Bellman 1957; Puterman 1994; Sutton & Barto 2018). In a survey, failing to anchor the foundations in the scholarly record is a major oversight.
3. **Non-Standard Writing Practices:** The paper fails to follow basic academic conventions for broad communication. For example, the core acronym **PAC (Probably Approximately Correct)** is used extensively in the title and abstract but is never given a grounded, narrative explanation or historical context in the introduction. It is expended only on page 8.
4. **Confusion Surrounding the "CSO" Framework:** The authors propose the CSO template to organize the field. The concept is interesting as a point of view for synthesizing the literature, if executed well. However, the authors fail to explicitly claim it as a contribution or motivate this specific decomposition. Presenting it as an a priori fact creates conceptual confusion. In addition, the mathematical notion of the concept is confusing at best.
5. **Reliance on Bullet Points vs. Narrative Synthesis:** Large portions of the paper (notably Section 13 and the Practitioner Toolkits) consist of exhaustive bulleted lists. This disrupts the reading flow and reinforces the impression of unpolished notes rather than a journal-quality manuscript.

**Resubmission Of Major Revision:**

The authors may consider submitting a major revision at a later time.